# Robust Model Selection and Nearly-Proper Learning for GMMs

**Allen Liu**
MIT
Cambridge, MA 02139
cliu568@mit.edu

**Jerry Li**
Microsoft Research
Redmond, WA 98052
jerrl@microsoft.com

**Ankur Moitra**
MIT
Cambridge, MA 02139
moitra@mit.edu

## Abstract

In learning theory, a standard assumption is that the data is generated from a finite mixture model. But what happens when the number of components is not known in advance? The problem of estimating the number of components, also called *model selection*, is important in its own right but there are essentially no known efficient algorithms with provable guarantees let alone ones that can tolerate adversarial corruptions. In this work, we study the problem of robust model selection for univariate Gaussian mixture models (GMMs). Given $\text{poly}(k/\epsilon)$ samples from a distribution that is $\epsilon$-close in TV distance to a GMM with $k$ components, we can construct a GMM with $\widetilde{O}(k)$ components that approximates the distribution to within $\widetilde{O}(\epsilon)$ in $\text{poly}(k/\epsilon)$ time. Thus we are able to approximately determine the minimum number of components needed to fit the distribution within a logarithmic factor. Prior to our work, the only known algorithms for learning arbitrary univariate GMMs either output significantly more than $k$ components (e.g. $k/\epsilon^2$ components for kernel density estimates) or run in time exponential in $k$. Moreover, by adapting our techniques we obtain similar results for reconstructing Fourier-sparse signals.

## 1 Introduction

Many works in learning theory operate under the assumption that the data is generated from a finite mixture model, and furthermore that the number of components is known in advance. But what happens when the number of components is not known in advance? The problem of estimating the number of components is called *model selection* and has been intensively studied in statistics for over fifty years [Neyman and Scott, 1966]. Indeed, in many scientific applications, it is the central issue. Consider the motivation given by Chen et al. [2004]: In genetics, we might have a continuous-valued trait, like height, that can be measured across a population and we want to understand its genetic basis. But is the underlying genetic mechanism simple or complex? Is it controlled by just a few genes or are there many more genes waiting to be discovered that each have a small effect on it?

From a statistical perspective, what makes model selection challenging is that the standard analysis of the likelihood ratio test breaks down because of lack of regularity and non-identifiability [Hartigan, 1985]. Despite many attempts [Ghosh and Sen, 1984, Lo et al., 2001, Huang et al., 2017] and rejoinders [Jeffries, 2003], even understanding the asymptotic distribution of the likelihood ratio statistics has remained a long-standing challenge in the field [Kasahara and Shimotsu, 2015]. From an algorithmic standpoint, the problem is even more difficult.

In this work, we study the problem of robust model selection for one-dimensional Gaussian mixture models with $k$ components ($k$-GMMs for short). A natural approach for this problem is via *agnostic proper learning*, where the task is to, given samples from an unknown distribution, output the best $k$-GMM approximation to this distribution in TV distance. An efficient agnostic proper learning

algorithm, combined with standard tools from hypothesis testing, would immediately yield an algorithm for model selection.

Unfortunately, while there are many efficient algorithms for learning one-dimensional GMMs, they all fall into one of several categories: (1) They assume some strong separation conditions on the components so that the samples can be clustered based on which component they were generated from. (2) They solve the harder problem of learning the parameters of the components, which information-theoretically requires the number of samples to be exponential in $k$ [Moitra and Valiant, 2010]. (3) They employ brute-force search [Daskalakis and Kamath, 2014, Acharya et al., 2014] or solve a system of polynomial inequalities [Li and Schmidt, 2017], and run in time exponential in $k$. (4) They learn an approximation that is either not a GMM, e.g. a piece-wise polynomial approximation [Chan et al., 2013, Acharya et al., 2017] or output a GMM where the number of components is much larger than $k$ [Wu and Xie, 2018, Devroye and Lugosi, 2012, Bhaskara et al., 2015]. (5) They assume that the components in the GMM have the same or similar variances and means not too far apart so that there is a good approximation to the density with just a logarithmic number of components [Wu and Yang, 2018, Polyanskiy and Wu, 2020]. In all cases, these guarantees are insufficient for efficient model selection, and/or yield a trivial approximation to the number of components in a GMM except in restricted settings. In this work, we ask: Are there efficient algorithms for learning arbitrary one-dimensional GMMs that output an approximation with $\widetilde{O}(k)$ components? Relatedly: Are there efficient algorithms for approximating the number of components in a GMM? We give efficient algorithms whose running time and sample complexity are polynomial in $k$ for both of these problems, and also the related problem of reconstructing Fourier-sparse signals with an unknown number of frequencies.

## 1.1 Learning and model selection for GMMs

Our main result is a new robust learning algorithm for one-dimensional GMMs. We show:

**Theorem 1.1.** *Let $k, \epsilon > 0$ be parameters and let $f$ be a distribution such that $d_{\mathsf{TV}}(\mathcal{M}, f) \leq \epsilon$ for some unknown mixture of Gaussians $\mathcal{M} = w_1 G_1 + \cdots + w_k G_k$. Assume that we are given $\widetilde{O}(k/\epsilon^2)$ samples from $f$. Then there is an algorithm that runs in $\mathrm{poly}(k/\epsilon)$ time and with probability $0.9$ (over the random samples), outputs a mixture of $\widetilde{O}(k)$ Gaussians, $\widetilde{\mathcal{M}}$, such that*

$$d_{\mathsf{TV}}(\widetilde{\mathcal{M}}, f) \leq \widetilde{O}(\epsilon) \,.$$

In contrast to other known learning algorithms (discussed earlier), our learning algorithm works for arbitrary GMMs, runs in polynomial time and uses a polynomial number of samples, and while it does not output a GMM with exactly $k$ components, it does the next best thing: it outputs a GMM with at most a polylogarithmic factor more components.

As a corollary, we also give an algorithm for robust approximate model selection for GMMs. The connection to model selection is that when our algorithm fails to find a GMM with $\widetilde{O}(k)$ components that fits the data we can be assured that there must more than $k$ components to begin with. Notice in particular that improper approximations by themselves do not suffice for the model selection problem, as a good improper approximation could exist even if the distribution is far from any GMM with $\widetilde{O}(k)$ components.

**Theorem 1.2.** *Let $k, \epsilon > 0$ be parameters we are given. Let $\mathcal{F}_1$ be the family of distributions that are $\epsilon$-close to a $k$-GMM with $k$ components (in TV distance). Let $\mathcal{F}_2$ be the family of distributions that are not $\widetilde{O}(\epsilon)$-close to any GMM with $\widetilde{O}(k)$ components. There is an algorithm that given $\mathrm{poly}(k/\epsilon)$ samples from a known distribution $\mathcal{D}$, runs in $\mathrm{poly}(k/\epsilon)$ time, and outputs 1 if $\mathcal{D} \in \mathcal{F}_1$ and outputs 2 if $\mathcal{D} \in \mathcal{F}_2$ both with failure probability at most $0.2$.*

**Remark.** *Even if the distribution $\mathcal{D}$ is completely unknown and we are only given samples from it, the above result still holds as long as $\mathcal{D}$ is somewhat well behaved (note that such an assumption is necessary as hypothesis testing with respect to total variation distance without any assumptions on $\mathcal{D}$ is impossible). In particular we can use piecewise polynomial approximation [Chan et al., 2013] or kernel density estimates [Terrell and Scott, 1992] to learn a distribution $\mathcal{D}'$ that is close to $\mathcal{D}$ that we have an explicit form for and then run the hypothesis test using $\mathcal{D}'$.*

## 1.2 Fourier sparse interpolation

Our techniques also immediately apply to the problem of Fourier sparse interpolation, where the goal is to interpolate a signal based on noisy measurements of it at a few points [Chen et al., 2016]. We say that a function $\mathcal{M}$ is $(k, C)$ simple if it can be written in the form

$$\mathcal{M}(t) = \sum_{j=1}^{k} a_j e^{2\pi i \theta_j t} \ ,$$

where additionally $\sum_j |a_j| \le C$. In other words, a function is $(k, C)$ simple if it is $k$-sparse in the Fourier domain, and its Fourier coefficients are bounded in $\ell_1$ by $C$.

We consider the following problem. We get query access to a function $f(t) = \mathcal{M}(t) + \eta(t)$ at any point in the interval $[-1, 1]$, where $\mathcal{M}$ is $(k, C)$ simple and has all frequencies in the interval $[-F, F]$, and $\eta(t)$ is noise that we will assume is bounded in $L_2$ norm. The goal is to compute a Fourier-sparse approximation $\widetilde{\mathcal{M}}(t)$ that is close to $f(t)$, in the sense that its error is comparable to that of $\mathcal{M}(t)$. Recently Chen et al. [2016] showed how to construct an approximation $\widetilde{\mathcal{M}}(t)$ that satisfies

$$\|f(t) - \widetilde{\mathcal{M}}(t)\|_2 \le \|\eta(t)\|_2 + \epsilon \|\mathcal{M}(t)\|_2$$

where the $L_2$ norm is taken over the interval $[-1, 1]$. Their algorithm works for any $\epsilon > 0$ and uses $\mathrm{poly}(k, \log 1/\epsilon) \log F$ measurements. Moreover the $\widetilde{\mathcal{M}}(t)$ that they output is $\mathrm{poly}(k, \log 1/\epsilon)$-Fourier sparse. Similarly to the GMM setting, a natural goal is to perform robust interpolation but with tighter bounds on the number of frequencies. We show:

**Theorem 1.3.** *Let $f, \mathcal{M}$ be as above where $\mathcal{M}$ is $(k, 1)$-simple. Then for any desired accuracy $\epsilon > 0$ and constant $c > 0$, in $\mathrm{poly}(k, \log 1/\epsilon) \log F$ queries and $\mathrm{poly}(k/c, \log 1/\epsilon) \log^2 F$ time, we can output a function $\widetilde{\mathcal{M}}$ such that with probability $1 - 2^{-\Omega(k)}$,*

1. *$\widetilde{\mathcal{M}}$ is $\widetilde{O}(k)$-Fourier sparse with $\|\widehat{\widetilde{\mathcal{M}}}\|_1 \le \widetilde{O}(k)$*

2. *$\int_{-1+c}^{1-c} |\widetilde{\mathcal{M}} - f|^2 \le \widetilde{O}\left(\epsilon^2 + \int_{-1}^{1} |f - \mathcal{M}|^2\right)$*

**Remark.** *Note the constraints $\|\widehat{\mathcal{M}}\|_1$ and $\|\widehat{\widetilde{\mathcal{M}}}\|_1$ translate into bounds on the sizes of the coefficients of the exponentials in $\mathcal{M}$ and $\widetilde{\mathcal{M}}$ respectively.*

The natural open question left by our work is to improve the sparsity bounds, both for interpolation/learning and model selection. In principle it could be possible that there are efficient algorithms for these problems, however it now seems somewhat unlikely. Even without noise, learning a Gaussian mixture model with $k$ components without a separation condition in time $\mathrm{poly}(k, 1/\epsilon)$ is open. From our work (see Section 2), we see that even in the well-conditioned case this is equivalent to finding a non-trivially sparse solution to a system of polynomial equations where there seems to be no structure that makes algorithmic search better than brute-force possible. Moreover, this question has already been open for many years, but there hasn't been any progress on proper learning. Thus, we conjecture that both the learning and model selection problems are computationally hard if we are not allowed to relax the number of components.

## 1.3 Related work

There is a vast literature on the three problems we consider. Here we will give a more detailed review of related work.

**Learning Mixtures of Gaussians and Model Selection** Since the pioneering work of Pearson [1894], mixtures of Gaussians have become one of the most ubiquitous and well-studied generative models in both theory and practice. Numerous problems have been studied on the context of learning mixtures of Gaussians, including clustering [Dasgupta, 1999, Vempala and Wang, 2004, Achlioptas and McSherry, 2005, Dasgupta and Schulman, 2007, Arora and Kale, 2007, Kumar and Kannan, 2010, Awasthi and Sheffet, 2012, Mixon et al., 2017, Hopkins and Li, 2018, Kothari et al., 2018, Diakonikolas et al., 2018], learning in the presence of adversarial noise in high dimensional

settings [Diakonikolas et al., 2018, Hopkins and Li, 2018, Kothari et al., 2018, Bakshi et al., 2020, Diakonikolas et al., 2020, Kane, 2021, Liu and Moitra, 2020, 2021], parameter estimation [Kalai et al., 2010, Belkin and Sinha, 2015, Moitra and Valiant, 2010, Hardt and Price, 2015], learning in smoothed settings [Hsu and Kakade, 2013, Anderson et al., 2014, Bhaskara et al., 2014, Ge et al., 2015], and density estimation [Devroye and Lugosi, 2012, Chan et al., 2014, Acharya et al., 2017].

Of particular interest to us is the line of work on proper learning [Feldman et al., 2006, Acharya et al., 2014, Li and Schmidt, 2017, Ashtiani et al., 2018], where the goal is to output a mixture of $k$-Gaussians which is close in total variation to the underlying ground truth. Unfortunately, while the sample complexity of these algorithms is usually polynomial, the runtime for all known approaches is exponential in $k$. In contrast, our runtimes are polynomial, albeit for a relaxed version of the problem, where the output is allowed to be a mixture of $k'$ Gaussians, for $k' > k$.

For this "semi-proper" regime, efficient algorithms are known, albeit either only for restricted settings, or with significantly worse quantitative results than we achieve. In the "well-conditioned" case, where the means are close together, and the variances of all the components are comparable, the aforementioned work of [Wu and Yang, 2018, Polyanskiy and Wu, 2020] demonstrates that the nonparametric MLE can efficiently obtain an estimate using only logarithmically many pieces. However, the nonparametric MLE is not suited for the general setting, where the means could be far apart, and variances could be very different, and will not converge in general. Moreover, while nonparametric MLE is robust to perturbations in KL, it is not robust to perturbations in total variation distance, as we consider here.

For the general case, by using kernel density estimates, one can achieve $\epsilon$ approximation using $k' = O(k/\epsilon^C)$ for some constant $C$ [Devroye and Lugosi, 2012]. Similarly Bhaskara et al. [2015] achieves $\epsilon$ error using $k' = O(k/\epsilon^3)$ pieces. That is, for both of these approaches, they require a number of pieces which scales polynomially with $1/\epsilon$. In comparison, our dependence on $\epsilon$ in terms of the number of pieces is logarithmic.

As discussed previously, there are strong connections between proper learning and model selection [Neyman and Scott, 1966, Hartigan, 1985, Ghosh and Sen, 1984, Lo et al., 2001, Jeffries, 2003, Kasahara and Shimotsu, 2015, Huang et al., 2017]. Related notions have been considered in distribution testing [Parnas et al., 2006, Valiant and Valiant, 2010a,b, 2011, Jiao et al., 2016, 2017, Han et al., 2016] and testing properties of boolean functions [Diakonikolas et al., 2007, Iyer et al., 2021].

**Continuous Time Sparse Fourier Transforms**   Sparse Fourier transforms in the continuous setting, also known as sparse Fourier transforms off the grid, has been the subject of intensive study. Indeed, the first algorithm for this problem dates back to Prony [1795]. Modern algorithms include MUSIC [Schmidt, 1982], ESPRIT [Roy et al., 1986], maximum likelihood estimators [Bresler and Macovski, 1986], convex programming based methods [Candès and Fernandez-Granda, 2014] and the matrix pencil method [Moitra, 2015].

Most of these works, especially those that work in a noisy setting, require a frequency gap. Moreover they require more than $k$ samples (their bound usually depends on the frequency gap), even if the underlying signal is $k$-sparse in the Fourier domain. A recent line of work has focused on the problem of improving the sample complexity – in particular getting bounds which only depend on $k$ with runtimes that are polynomial in $k$ [Fannjiang and Liao, 2012, Duarte and Baraniuk, 2013, Tang et al., 2013, 2014, Boufounos et al., 2015, Huang and Kakade, 2015, Price and Song, 2015]. The setting where there is no gap and there is noise is particularly challenging. One approach is to relax the definition of a frequency gap, and require it only between "clusters" of frequencies [Batenkov et al., 2020]. Another line of work [Avron et al., 2019, Chen and Price, 2019] shows how to output a hypothesis which is $k$-sparse without any gap assumptions and with sample complexity which is polynomial in $k$. However these methods run in exponential time. As we previously discussed, the most relevant works to us are Chen et al. [2016] and Chen and Price [2019], which give an algorithm whose running time and sample complexity are polynomial in $k$ that works without any gap assumptions, but for a relaxation where we are allowed to output a $\widetilde{O}(k^2)$-Fourier sparse signal.

## 2   Technical overview

We now give an overview of our approach. We will focus on just the GMM case in this overview. Our approach for sparse Fourier interpolation follows a very similar outline. We first present our

techniques assuming that we have explicit access to $f$. In Section 2.3 we show how to reduce to this case when we are only given samples. In other words, the problem is as follows: we are given a function $f$, and we want to find a sparse approximation to $f$ as a nonnegative sum of Gaussians, i.e. we want to write

$$f \sim a_1 G_1 + \cdots + a_n G_n$$

with $n$ small, where each $G_i$ is a Gaussian.

## 2.1 Well-conditioned case

We first solve the "well-conditioned" case. Roughly, we say that a GMM is well-conditioned if the variances of the components are all constant scale and the means are all not too far from zero. Formally, we have the following definition:

**Definition 2.1.** *We say a Gaussian $G = N(\mu, \sigma^2)$ is $\delta$-well-conditioned if $|\mu| \leq \delta$ and $|\sigma^2 - 1| \leq \delta$. Furthermore we say a mixture of Gaussians $\mathcal{M} = w_1 G_1 + \cdots + w_k G_k$ is $\delta$-well-conditioned if all of the components $G_1, \ldots, G_k$ are $\delta$-well-conditioned.*

Naturally, our techniques also apply to a shared scaling and/or translation of the components, but we will ignore this for now. Earlier work of [Wu and Yang, 2018, Polyanskiy and Wu, 2020] proved an important structural result that a well-conditioned GMM can be $\epsilon$-approximated by a mixture with $O(\log 1/\epsilon)$ components. However we will want a robust and algorithmic version: In particular, instead of requiring the distribution to be exactly a well-conditioned GMM, we will only require that it be close in total variation distance. Even in this setting, with some level of model misspecification, we want an efficient algorithm for constructing an approximating GMM with few components. To this end, a key result, proved in Section A, is:

**Lemma 2.2.** *Let $\epsilon > 0$ be a parameter. Assume we are given access to a distribution $f$ such that $d_{TV}(f, \mathcal{M}) \leq \epsilon$ where $\mathcal{M} = w_1 G_1 + \cdots + w_k G_k$ is a $0.5$-well-conditioned mixture of Gaussians. Then we can compute, in $\mathrm{poly}(1/\epsilon)$ time, a mixture $\widetilde{\mathcal{M}}$ of at most $O(\log 1/\epsilon)$ Gaussians such that $d_{TV}(\mathcal{M}, \widetilde{\mathcal{M}}) \leq \widetilde{O}(\epsilon)$.*

Our approach departs from the moment matching framework of Wu and Yang [2018], Polyanskiy and Wu [2020]. Instead we take the probability density function of any well-conditioned Gaussian $G_j$. We can expand it as a Taylor series around $0$ of the form

$$G_j(x) = c_{G_j}^{(0)} + \frac{c_{G_j}^{(1)} x}{1!} + \frac{c_{G_j}^{(2)} x^2}{2!} + \ldots$$

for some coefficients $c_{G_j}^{(i)}$. We can then associate it with the vector $c_{G_j} = (c_{G_j}^{(0)}, \ldots, c_{G_j}^{(\ell-1)})$ of length $\ell = O(\log 1/\epsilon)$. Then, for any well-conditioned mixture $\mathcal{M} = a_1 G_1 + \cdots + a_k G_n$, we can associate it with the corresponding convex combination of the vectors of its components, i.e., we define $c_{\mathcal{M}} = a_1 c_{G_1} + \ldots + \ldots a_k c_{G_k} \in \mathbb{R}^\ell$.

The point of this is the following implication: if two well-conditioned mixtures get mapped to vectors which are close, then these two mixtures must be close in total variation distance. The intuition is that when we write down the $L_1$ distance between the two mixtures, because the Taylor coefficients of Gaussians decay exponentially fast, the contribution of terms with degree $l > O(\log 1/\epsilon)$ to the integral becomes negligible.

Now, we can associate the set of well-conditioned mixtures with a convex body in $O(\log 1/\epsilon)$-dimensions, where the vertices of the convex body are given by single Gaussians. Consequently we can use Caratheodory's theorem to argue that any point within this body can be approximated as an $O(\log 1/\epsilon)$-sparse convex combination of the vertices, or equivalently, any well-conditioned mixture can we approximated by a mixture of $O(\log 1/\epsilon)$ well-conditioned Gaussians.

It remains to demonstrate how to actually find this sparse mixture of Gaussians. Naively, the number of vertices is infinite, as there are infinitely many well-conditioned Gaussians. However, it is not too hard to show that if we consider a slight coarsening of this body by only taking the vertices to be the vectors associated to the well-conditioned Gaussians which belong in some $\mathrm{poly}(1/\epsilon)$-sized net, then the quality of our solution only degrades by constant multiplicative factors. At this point we can appeal to standard results in convex optimization to find the desired sparse approximation. We defer the details of this argument to Section A.

## 2.2 Localization

After solving the well-conditioned case, the next step is to reduce the general case to the well-conditioned case via localization. We begin with an important definition.

**Definition 2.3** (Gaussian Multiplier). *For parameters $\mu, \sigma$, we define*

$$M_{\mu,\sigma^2}(x) = e^{-\frac{(x-\mu)^2}{2\sigma^2}}$$

*i.e. it is a Gaussian scaled so that its maximum value is 1.*

Gaussian multipliers will be crucial in the localization step. Now assume that $f$ can be written as some unknown $k$-sparse combination, say

$$f = a_1 G_1 + \cdots + a_k G_k$$

We can then modify $f$, e.g. by multiplying by a Gaussian multiplier $M_{\mu,\sigma^2}$. Heuristically, this operation changes the coefficients $a_1, \ldots, a_k$ in a predictable way. Namely, the coefficients $a_j$ of Gaussians $G_j$ that are far from $N(\mu, \sigma^2)$ are exponentially attenuated based on the distance to $N(\mu, \sigma^2)$. This effectively "localizes" the mixture. More formally,

**Claim 2.4.** *We have the identity*

$$M_{\mu,\sigma^2}(x) N(\mu_1, \sigma_1^2) = \frac{1}{\sqrt{1 + \frac{\sigma_1^2}{\sigma^2}}} e^{-\frac{(\mu_1-\mu)^2}{2(\sigma_1^2+\sigma^2)}} N\left(\frac{\mu\sigma_1^2 + \mu_1\sigma^2}{\sigma_1^2 + \sigma^2}, \frac{\sigma_1^2\sigma^2}{\sigma_1^2 + \sigma^2}\right).$$

*Proof.* We prove the above through direct computation.

$$M_{\mu,\sigma^2}(x) G_1(x) = e^{-\frac{(x-\mu)^2}{2\sigma^2} - \frac{(x-\mu_1)^2}{2\sigma_1^2}} \frac{1}{\sigma_1\sqrt{2\pi}} = \frac{1}{\sigma_1\sqrt{2\pi}} \cdot e^{-\frac{1}{2}\left(\left(\frac{1}{\sigma^2}+\frac{1}{\sigma_1^2}\right)x^2 - 2\left(\frac{\mu}{\sigma^2}+\frac{\mu_1}{\sigma_1^2}\right)x + \frac{\mu^2}{\sigma^2}+\frac{\mu_1^2}{\sigma_1^2}\right)}$$

$$= \frac{1}{\sigma_1\sqrt{2\pi}} \cdot \exp\left(-\frac{1}{2}\left(\sqrt{\frac{1}{\sigma^2}+\frac{1}{\sigma_1^2}} x - \frac{\frac{\mu}{\sigma^2}+\frac{\mu_1}{\sigma_1^2}}{\sqrt{\frac{1}{\sigma^2}+\frac{1}{\sigma_1^2}}}\right)^2 - \frac{1}{2}\cdot\frac{(\mu_1-\mu)^2}{\sigma_1^2+\sigma^2}\right)$$

$$= \frac{1}{\sqrt{1+\frac{\sigma_1^2}{\sigma^2}}} e^{-\frac{(\mu_1-\mu)^2}{2(\sigma_1^2+\sigma^2)}} N\left(\frac{\mu\sigma_1^2 + \mu_1\sigma^2}{\sigma_1^2 + \sigma^2}, \frac{\sigma_1^2\sigma^2}{\sigma_1^2 + \sigma^2}\right).$$

∎

The hope is that this will leave us with only components that are not too far from each other – exactly the well-conditioned case which we already know how to solve. If the variances of all of the components are comparable, then this is indeed the case. However, additional complications arise when one of the components $G_i = N(\mu, \sigma_i^2)$ has variance $\sigma_i \ll \sigma$ because this component will still have much smaller variance than the others after localizing. Nevertheless, we show that we can carefully localize at different scales, using smaller variance Gaussian multipliers to localize around smaller variance components so that all of the localized mixtures are well-conditioned.

The main remaining question is to select a good family of localizations so that we can then fully reconstruct the original mixture from the localized mixtures. Each localized mixture will cost us $O(\log 1/\epsilon)$ components, and therefore we must use at most $\widetilde{O}(k)$ different localizations. When all of the variances of the Gaussians are not too dissimilar, we can do so by leveraging the following structural result, which states that one can $\epsilon$-approximate the constant function using a sum of evenly spaced Gaussians with variance 1 and spacing $(\log 1/\epsilon)^{-1/2}$ (or smaller). The intuition behind this observation is that the Fourier transform of a Gaussian is also a Gaussian, which has exponential tail decay.

**Lemma 2.5.** *Let $0 < \epsilon < 0.1$ be a parameter. Let $c$ be a real number such that $0 < c \leq (\log 1/\epsilon)^{-1/2}$. Define*

$$f(x) = \sum_{j=-\infty}^{\infty} \frac{c}{\sqrt{2\pi}} M_{cj\sigma,\sigma^2}(x).$$

*Then $1 - \epsilon \leq f(x) \leq 1 + \epsilon$ for all $x$.*

*Proof.* WLOG $\sigma = 1$. Now the function $f$ is $c$-periodic and even, so we may consider its Fourier expansion

$$f(x) = a_0 + 2a_1 \cos\left(\frac{2\pi x}{c}\right) + 2a_2 \cos\left(\frac{4\pi x}{c}\right) + \dots$$

and we will now compute the Fourier coefficients. First note that

$$a_0 = \frac{1}{c} \int_0^c f(x) dx = \frac{1}{\sqrt{2\pi}} \sum_{j=-\infty}^{\infty} \int_{c(j+1)}^{cj} M_{0,1}(x) dx = 1 .$$

Next, for any $j \geq 1$,

$$a_j = \frac{1}{c} \int_0^c f(x) \cos\left(\frac{2\pi j x}{c}\right) dx = \frac{1}{\sqrt{2\pi}} \sum_{j=-\infty}^{\infty} \int_{c(j+1)}^{cj} M_{0,1}(x) \cos\left(\frac{2\pi j x}{c}\right) dx$$

$$= \frac{1}{\sqrt{2\pi}} \int_{-\infty}^{\infty} \frac{1}{2} \left( e^{\frac{-x^2}{2} + \frac{2\pi i j x}{c}} + e^{\frac{-x^2}{2} - \frac{2\pi i j x}{c}} \right) dx = e^{-\frac{2\pi^2 j^2}{c^2}}$$

where in the above we use the notation $i = \sqrt{-1}$. Using the assumption that $c \leq (\log 1/\epsilon)^{-1/2}$, it is clear that

$$\sum_{j=1}^{\infty} e^{-\frac{2\pi^2 j^2}{c^2}} \leq \frac{\epsilon}{2}$$

so we deduce that for any $x$,

$$|f(x) - 1| \leq 2(|a_1| + |a_2| + \cdots) = 2\sum_{j=1}^{\infty} e^{-\frac{2\pi^2 j^2}{c^2}} \leq \epsilon .$$

In other words, the function $f$ is between $1 - \epsilon$ and $1 + \epsilon$ everywhere and we are done. ∎

In light of the above lemma, we can use a set of evenly spaced Gaussian multipliers and simply sum the different localized mixtures. Note that it suffices to use $\widetilde{O}(k)$ different localizations because we only need to sum over the Gaussian multipliers that have some nontrivial overlap with one of the $k$ true components (since for Gaussian multipliers that are far from all of the components, the localized mixture will be approximately 0).

To handle the fully general case, when the variances of the Gaussians are unbounded, we need a generalization of the previous lemma that allows us to $\epsilon$-approximate the indicator function of an interval with a sum of $O(\log^2 1/\epsilon)$ Gaussians. The proof of this generalization is in Section B.

**Definition 2.6** (Significant Interval). *For a Gaussian multiplier $M_{\mu,\sigma^2}$, we say the $C$-significant interval of $M$ is $[\mu - C\sigma, \mu + C\sigma]$. We will use the same terminology for a Gaussian $N(\mu, \sigma^2)$.*

**Theorem 2.7.** *Let $l$ be a positive real number and $0 < \epsilon < 0.1$ be a parameter. There is a function $f$ with the following properties*

1. *$f$ can be written a linear combination of Gaussian multipliers*

$$f(x) = w_1 M_{\mu_1, \sigma_1^2}(x) + \cdots + w_n M_{\mu_n, \sigma_n^2}(x)$$

   *where $n = O(\log^2 1/\epsilon)$ and $0 \leq w_1, \dots, w_n \leq 1$*

2. *The $10\sqrt{\log 1/\epsilon}$-significant intervals of all of the $M_{\mu_i, \sigma_i^2}$ are contained in the interval $[-(1+\epsilon)l, (1+\epsilon)l]$*

3. *$0 \leq f(x) \leq 1 + \epsilon$ for all $x$*

4. *$1 - \epsilon \leq f(x) \leq 1 + \epsilon$ for all $x$ in the interval $[-l, l]$*

5. *$0 \leq f(x) \leq \epsilon$ for $x \geq (1+\epsilon)l$ and $x \leq -(1+\epsilon)l$*

We combine this structural result with a dynamic program which allows us to efficiently choose the scales at which to localize. Putting all of these pieces together yields our full algorithm, assuming we have access to the pdf of the unknown function. We show how to eliminate the need for pdf access below and present our full algorithm in complete detail in Section C.

## 2.3 Abstracting away the samples

In the previous sections, we have assumed that we have access to the underlying pdf function $f$. Typically, however, we only have sample access to the unknown distribution. To rectify this, we will use the improper learner in [Chan et al., 2013] (see Theorem 37) whose output is a piecewise polynomial. We can then only work with this piecewise polynomial, which is an explicit function that we can then perform explicit computations with.

**Definition 2.8.** *A function $f$ is $t$-piecewise degree $d$ if there is a partition of the real line into intervals $I_1, \ldots, I_t$ and polynomials $q_1(x), \ldots, q_t(x)$ of degree at most $d$ such that for all $i \in [t]$, $f(x) = q_i(x)$ on the interval $I_i$.*

The work in [Chan et al., 2013] guarantees to learn a piecewise polynomial $f'$ that is close to $\mathcal{M}$ in $L^1$ distance when given $\widetilde{O}(k/\epsilon^2)$ samples (and they also show that this sample complexity is essentially optimal).

**Theorem 2.9** (Chan et al. [2013])**.** *Let $\mathcal{M} = w_1 G_1 + \cdots + w_k G_k$ be an unknown mixture of Gaussians and $f$ a distribution such that $d_{\mathsf{TV}}(f, \mathcal{M}) \leq \epsilon$. There is an algorithm that, given $\widetilde{O}(k/\epsilon^2)$ samples from $f$, runs in $\mathrm{poly}(k/\epsilon)$ time and returns an $O(k)$-piecewise degree $O(\log 1/\epsilon)$ function $f'$ such that with $0.9$ probability (over the random samples),*

$$\|f' - f\|_1 \leq O(\epsilon) \,.$$

For technical reasons, we will need a few simple post-processing steps after using Theorem 2.9. We can ensure that the output hypothesis $f'$ is always nonnegative by splitting each polynomial into positive and negative parts and zeroing out the negative parts (since this will not increase the $L^1$ error). Finally, we can re-normalize so that the output $f'$ is actually a distribution. This renormalization at most doubles the $L^1$ error. Thus we have:

**Corollary 2.10.** *Let $\mathcal{M} = w_1 G_1 + \cdots + w_k G_k$ be an unknown mixture of Gaussians and $f$ a distribution such that $d_{\mathsf{TV}}(f, \mathcal{M}) \leq \epsilon$. There is an algorithm that, given $\widetilde{O}(k/\epsilon^2)$ samples from $\mathcal{D}$, runs in $\mathrm{poly}(k/\epsilon)$ time and returns an $O(k \log 1/\epsilon)$-piecewise degree $O(\log 1/\epsilon)$ function $f'$ such that $f'$ is a distribution and with $0.9$ probability (over the random samples),*

$$d_{\mathsf{TV}}(f, f') \leq O(\epsilon) \,.$$

## 2.4 Hypothesis testing for model selection

We now show how our result for model order selection, Theorem 1.2, follows immediately from combining Theorem 1.1 with a standard procedure for testing the TV-distance between two distributions from samples (see Yatracos [1985]).

**Claim 2.11.** *Let $\mathcal{D}_1, \mathcal{D}_2$ be two distributions for which we have explicitly computable density functions. Let $\epsilon, \tau > 0$ be parameters. Assume that we are given $O(1/\epsilon^2 \cdot \log 1/\tau)$ samples from $\mathcal{D}_1$ and can efficiently sample from $\mathcal{D}_2$. Then in $\mathrm{poly}(1/\epsilon \log 1/\tau)$ time, we can compute $d$ such that with probability $1 - \tau$,*

$$|d - d_{\mathsf{TV}}(\mathcal{D}_1, \mathcal{D}_2)| \leq \epsilon \,.$$

*Proof of Theorem 1.2.* We can run the algorithm in Theorem 1.1 with parameters $k, \epsilon$ to obtain an output distribution $\widetilde{\mathcal{M}}$ that is a mixture of $\widetilde{O}(k)$ Gaussians. We can then use Claim 2.11 with parameters $\epsilon, 0.01$ to measure the TV-distance between $\widetilde{\mathcal{M}}$ and $\mathcal{D}$ (note that we have explicit access to the pdf of $\mathcal{D}$) and output 1 or 2 depending on if our estimate of the TV distance is less than $\widetilde{O}(\epsilon)$. Combining the guarantees of Theorem 1.1 and Claim 2.11 ensures that our output satisfies the desired properties. ∎

## 2.5 Sparse Fourier

We now briefly describe how our techniques can be used for sparse Fourier reconstruction. Recall that the problem is to, given query access to a function $f$ on $[-1, 1]$ which is approximately $k$-Fourier sparse, approximate it with an $\widetilde{O}(k)$-Fourier sparse function. As before, we first abstract away the query access, by leveraging the following result from Chen et al. [2016]:

**Theorem 2.12** (Theorem 1.1 in Chen et al. [2016]). *Let $f$ be a function defined on $[-1, 1]$ and assume we are given query access to $f$. Let $\mathcal{M}$ be a function that is $(k, 1)$-simple and has frequencies in the interval $[-F, F]$. Then for any desired accuracy $\epsilon$, in $\mathrm{poly}(k, \log 1/\epsilon) \log F$ samples and $\mathrm{poly}(k, \log 1/\epsilon) \log^2 F$ time, we can output a function $f'$ such that with probability $1 - 2^{-\Omega(k)}$,*

1. *$f'$ is $(\mathrm{poly}(k, \log 1/\epsilon), \exp(\mathrm{poly}(k, \log 1/\epsilon)))$- simple*

2.

$$\int_{-1}^{1} |f' - f|^2 \leq O\left(\epsilon^2 + \int_{-1}^{1} |f - \mathcal{M}|^2\right).$$

**Remark.** *While the bound on the coefficients of $f'$ is not explicitly stated in Theorem 1.1 in Chen et al. [2016], it immediately follows from the proof.*

Our algorithm for postprocessing this into a $\widetilde{O}(k)$-Fourier sparse signal follows roughly the same steps as in the Gaussian case. First, we show that in a certain "well-conditioned" regime, namely, when the frequencies are not too dissimilar, there is a signal using $O(\log 1/\epsilon)$ frequencies which approximates the function. To handle the general case, we use localizations based on carefully chosen kernels to reduce every signal to a sum of well-conditioned signals (at least, approximately).

One important distinction between the GMM and sparse Fourier reconstruction setting we highlight is that in the latter, the goal is usually to have runtimes which scale logarithmically with $1/\epsilon$, whereas in the GMM setting, $\mathrm{poly}(1/\epsilon)$ sample complexity and thus runtime is unavoidable. However, our naive method of solving the well-conditioned case required constructing a net of $\mathrm{poly}(1/\epsilon)$ many Gaussians, and thus required $\mathrm{poly}(1/\epsilon)$ runtime. To circumvent this difficulty, we demonstrate that in fact this can be improved, and that by being more careful, and choosing the (much smaller) set of vertices based on the Chebyshev points, we can in fact improve this runtime significantly. See Sections D and E for a full treatment of our algorithm.

### 2.6   Paper organization

The remainder of the paper will be devoted to proving Theorem 1.1, our main result for GMMs and Theorem 1.3, our main result for sparse Fourier reconstruction. Due to space constraints, the remaining parts are deferred to the appendix. We first present the proof of our result for GMMs. In Section A, we deal with the well-conditioned case. In Section B, we present some tools for localization which we will then use in Section C to prove our full result for GMMs. We then present the proof of our main result for sparse Fourier reconstruction which follows a very similar outline. We deal with the well-conditioned case in Section D and then the general case in Section E. Appendix F contains several basic tools that will be used throughout the paper.

## Acknowledgments and Disclosure of Funding

AL was supported in part by an NSF Graduate Research Fellowship and a Fannie and John Hertz Foundation Fellowship. AM was supported in part by a Microsoft Trustworthy AI Grant, NSF CAREER Award CCF-1453261, NSF Large CCF1565235, a David and Lucile Packard Fellowship and an ONR Young Investigator Award.

## 3   Ethics and broader impact

Our work is purely theoretical and we do not think there are any ethical issues or potential negative societal impacts.

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
