## Supplementary Materials

## A   Well-Conditioned Case: Learning GMMs

We now deal with learning well-conditioned GMMs. We begin by formally specifying the properties that we want the components of the mixture to have. Roughly, we want the components to have comparable variances and the separation between their means cannot be too large compared to the variances. This means that after applying a suitable linear transformation, the components are all not too far from the standard Gaussian $N(0, 1)$.

**Definition A.1.** *We say a Gaussian where $G = N(\mu, \sigma^2)$ is $\delta$-well-conditioned if*

- $|\mu| \leq \delta$
- $|\sigma^2 - 1| \leq \delta$

*We say a mixture of Gaussians $\mathcal{M} = w_1 G_1 + \cdots + w_k G_k$ is $\delta$-well-conditioned if all of the components $G_1, \ldots, G_k$ are $\delta$-well-conditioned.*

We now state our learning result for well-conditioned mixtures.

**Lemma A.2.** *Let $\epsilon > 0$ be a parameter. Assume we are given access to a distribution $f$ such that $d_{\mathsf{TV}}(f, \mathcal{M}) \leq \epsilon$ where $\mathcal{M} = w_1 G_1 + \cdots + w_k G_k$ is a 0.5-well-conditioned mixture of Gaussians. Then we can compute, in $\mathrm{poly}(1/\epsilon)$ time, a mixture $\widetilde{\mathcal{M}}$ of at most $O(\log 1/\epsilon)$ Gaussians such that $d_{\mathsf{TV}}(\mathcal{M}, \widetilde{\mathcal{M}}) \leq \widetilde{O}(\epsilon)$.*

**Remark.** *Note that in the well-conditioned case, the number of components in the mixture that we compute does not depend on $k$.*

Our algorithm for proving Lemma A.2 can be broken down into two parts. In the first part, we find a mixture of $\mathrm{poly}(1/\epsilon)$ Gaussians that approximates $f$. We then show how to reduce this mixture of $\mathrm{poly}(1/\epsilon)$ Gaussians to $O(\log 1/\epsilon)$ Gaussians by using the Taylor series approximation to a Gaussian.

**Lemma A.3.** *Let $\epsilon > 0$ be a parameter. Assume we are given access to a distribution $f$ such that $d_{\mathsf{TV}}(f, \mathcal{M}) \leq \epsilon$ where $\mathcal{M} = w_1 G_1 + \cdots + w_k G_k$ is a 0.5-well-conditioned mixture of Gaussians. Then we can compute, in $\mathrm{poly}(1/\epsilon)$ time, a mixture of at most $O(1/\epsilon^2)$ Gaussians that is $\widetilde{O}(\epsilon)$-close to $\mathcal{M}$ in TV distance.*

*Proof.* First, let $\mathcal{T}$ be the set of all 0.5-well-conditioned Gaussians such that $\mu$ and $\sigma^2$ are integer multiples of $0.1\epsilon$. Note $|\mathcal{T}| = O(1/\epsilon^2)$.

By rounding all of the Gaussians $G_1, \ldots, G_k$ to the nearest element of $\mathcal{T}$ (this increases our $L^1$ error by at most $\epsilon$), we may assume that all of the components $G_1, \ldots, G_k$ are actually in $\mathcal{T}$. Now note that since $\|f - w_1 G_1 - \cdots - w_k G_k\|_1 \leq 2\epsilon$, we have for all $x$,

$$|\widehat{f}(x) - w_1 \widehat{G_1}(x) - \cdots - w_k \widehat{G_k}(x)| \leq 2\epsilon \tag{1}$$

where $\widehat{G_j}$ denotes taking the Fourier transform of the pdf of the Gaussian $G_j$. Let $l = \lceil \log 1/\epsilon \rceil$. We now have,

$$\int_{-l}^{l} |\widehat{f}(x) - w_1 \widehat{G_1}(x) - \cdots - w_k \widehat{G_k}(x)|^2 dx \leq O(l\epsilon^2).$$

Now let all of the Gaussians in $\mathcal{T}$ be $G_1, \ldots, G_m$ where $m = |\mathcal{T}|$. By Lemma F.10 (and splitting into real and imaginary parts), we can compute in $\mathrm{poly}(1/\epsilon)$ time, nonnegative weights $\widetilde{w_1}, \ldots, \widetilde{w_m}$ with $\widetilde{w_1} + \cdots + \widetilde{w_m} \leq 1$ such that

$$\int_{-l}^{l} |\widehat{f}(x) - \widetilde{w_1} \widehat{G_1}(x) - \cdots - \widetilde{w_m} \widehat{G_m}(x)|^2 dx \leq O(l\epsilon^2)$$

which by Cauchy Schwarz implies that

$$\int_{-l}^{l} |\widehat{f}(x) - \widetilde{w_1} \widehat{G_1}(x) - \cdots - \widetilde{w_m} \widehat{G_m}(x)| dx \leq O(l\epsilon).$$

Now note that since all of the Gaussians $G_1, \ldots, G_m$ are $0.5$-well-conditioned, their Fourier transforms $\widehat{G_j}$ also decay rapidly away from $[-l, l]$ so combining the above with (1), we deduce that

$$\int_{-\infty}^{\infty} |(\widetilde{w_1}\widehat{G_1}(x)) - \cdots - \widetilde{w_m}\widehat{G_m}(x)) - w_1\widehat{G_1}(x) - \cdots - w_k\widehat{G_k}(x)| \leq O(l\epsilon) \,.$$

From the Fourier transform of the above we then get for all $x$

$$|\widetilde{w_1}G_1(x) + \cdots + \widetilde{w_m}G_m(x) - w_1G_1(x) - \cdots - w_mG_m(x)| \leq O(l\epsilon)$$

and since all of the Gaussians involved are $0.5$-well-conditioned, they all decay rapidly outside the interval $[-l, l]$ and we conclude

$$\int_{-\infty}^{\infty} |\widetilde{w_1}G_1(x) + \cdots + \widetilde{w_m}G_m(x) - w_1G_1(x) - \cdots - w_mG_m(x)|dx \leq O(l^2\epsilon) \,.$$

Finally, note that by the above, we must have $1 - O(l^2\epsilon) \leq \widetilde{w_1} + \cdots + \widetilde{w_m} \leq 1 + O(l^2\epsilon)$ so rescaling to an actual mixture i.e. so that the weights $\widetilde{w_1} + \cdots + \widetilde{w_m} = 1$, will affect the above error by at most $O(l^2\epsilon)$. Thus, we can output this mixture and we are done. ∎

Next, as an immediate consequence of Lemma F.8, a $0.5$-well-conditioned Gaussian can be well approximated by its Taylor expansion.

**Corollary A.4.** *Let $G = N(\mu, \sigma^2)$ be a $0.5$-well-conditioned Gaussian. Let $\epsilon > 0$ be some parameter and let $l = \lceil \log 1/\epsilon \rceil$. Then we can compute a polynomial $P_G(x)$ of degree $(10l)^2$ such that for all $x \in [-l, l]$,*

$$|G(x) - P_G(x)| \leq O(\epsilon) \,.$$

*Proof.* This follows immediately from using Lemma F.8 and applying the appropriate linear transformation to the polynomial. ∎

We can now complete the proof of Lemma A.2 by using Lemma A.3 and then using Corollary A.4 and Caratheodory to reduce the number of components.

*Proof of Lemma A.2.* By Lemma A.3, we can compute a mixture

$$\widetilde{\mathcal{M}} = \widetilde{w_1}\widetilde{G_1} + \cdots + \widetilde{w_m}\widetilde{G_m}$$

such that $m = O(1/\epsilon^2)$ and

$$\|\widetilde{\mathcal{M}} - \mathcal{M}\|_1 \leq \widetilde{O}(\epsilon) \,.$$

For each Gaussian $\widetilde{G_j}$, let $P_{\widetilde{G_j}}(x)$ be the polynomial computed in Lemma F.8. Write

$$P_{\widetilde{G_j}}(x) = a_{j,0} + a_{j,1}x + \cdots + a_{j,(10l)^2}x^{(10l)^2} \,.$$

Define the vector

$$v_j = (a_{j,0}, a_{j,1}, \ldots, a_{j,(10l)^2}) \,.$$

Now the point $\widetilde{w_1}v_1 + \cdots + \widetilde{w_m}v_m$ is in the convex hull of $v_1, \ldots, v_m$. By Caratheodory (since the space is $(10l)^2 + 1$-dimensional), it must be in the convex hull of some $(10l)^2 + 1$ of the vertices. Thus, we can compute indices $i_0, \ldots, i_{(10l)^2}$ and nonnegative weights $w'_0, \ldots, w'_{(10l)^2}$ summing to 1 such that

$$\widetilde{w_1}v_1 + \cdots + \widetilde{w_m}v_m = w'_0 v_{i_0} + \cdots + w'_{(10l)^2}v_{i_{(10l)^2}} \,.$$

The above implies that for all $x$,

$$\widetilde{w_1}P_{\widetilde{G_1}}(x) + \cdots + \widetilde{w_m}P_{\widetilde{G_m}}(x) = w'_0 P_{\widetilde{G_{i_0}}}(x) + \cdots + w'_{(10l)^2}P_{\widetilde{G_{i_{(10l)^2}}}}(x) \,.$$

Now by Corollary A.4 and the fact that all of the Gaussians are $0.5$-well-conditioned, meaning that they decay rapidly outside of $[-l, l]$, we conclude that if we set

$$\mathcal{M}' = w'_0\widetilde{G_{i_0}} + \cdots + w'_{(10l)^2}\widetilde{G_{i_{(10l)^2}}}$$

then

$$\|\widetilde{\mathcal{M}} - \mathcal{M}'\|_1 \leq \widetilde{O}(\epsilon)$$

and then we have

$$\|\mathcal{M} - \mathcal{M}'\|_1 \leq \|\widetilde{\mathcal{M}} - \mathcal{M}\|_1 + \|\widetilde{\mathcal{M}} - \mathcal{M}'\|_1 \leq \widetilde{O}(\epsilon)$$

as desired. ∎

We can slightly improve Lemma A.2 to work even when we do not have a precise estimate of $d_{\mathsf{TV}}(f, \mathcal{M})$ since we can just repeatedly decrease our target accuracy until we cannot improve our accuracy further. Recall that we can use Claim 2.11 to test the $L^1$ distance between two distributions. We now have the following (slight) improvement of Lemma A.2.

**Corollary A.5.** *Let $\epsilon > 0$ be a parameter. Let $\mathcal{M} = w_1 G_1 + \cdots + w_k G_k$ be an unknown 0.5-well-conditioned mixture of Gaussians. Assume we are given access to a distribution $f$. Then we can compute, in $\mathrm{poly}(1/\epsilon)$ time, a mixture $\widetilde{\mathcal{M}}$ of at most $O(\log 1/\epsilon)$ Gaussians such that with high probability,*

$$d_{\mathsf{TV}}(f, \widetilde{\mathcal{M}}) \leq \epsilon^2 + \mathrm{poly}(\log 1/\epsilon) d_{\mathsf{TV}}(f, \mathcal{M}).$$

*Proof.* We can simply start from $\epsilon' = 1$ and run the algorithm in Lemma A.2 with parameter $\epsilon'$ and then estimate $d_{\mathsf{TV}}(f, \widetilde{\mathcal{M}})$ using Claim 2.11. If $d_{\mathsf{TV}}(f, \widetilde{\mathcal{M}}) \leq \epsilon' \mathrm{poly}(\log 1/\epsilon)$ then we can decrease $\epsilon'$ by a factor of 0.9 and repeat. Repeating this process and taking the smallest accuracy $\epsilon' \geq \epsilon^3$ for which the above check succeeds, we get (from the guarantee of Lemma A.2) that

$$d_{\mathsf{TV}}(f, \widetilde{\mathcal{M}}) \leq \epsilon^2 + \mathrm{poly}(\log 1/\epsilon) d_{\mathsf{TV}}(f, \mathcal{M})$$

and we are done. ∎

# B Function Approximations Using Gaussians

In this section, we present several results about approximating functions as a sum of Gaussians. These results will be key building blocks in the localization steps of both of our algorithms. The main result of this section, Theorem B.3, allows us to $\epsilon$-approximate the indicator function of an interval as a sum of $\mathrm{poly}(\log 1/\epsilon)$-Gaussians.

First, it will be convenient to renormalize Gaussians so that their maximum value is 1. After renormalization, we call them Gaussian multipliers.

**Definition B.1** (Gaussian Multiplier). *For parameters $\mu, \sigma$, we define*

$$M_{\mu, \sigma^2}(x) = e^{-\frac{(x-\mu)^2}{2\sigma^2}}$$

*i.e. it is a Gaussian scaled so that its maximum value is 1.*

We also introduce the some additional terminology.

**Definition B.2** (Significant Interval). *For a Gaussian multiplier $M_{\mu, \sigma^2}$, we say the $C$-significant interval of $M$ is $[\mu - C\sigma, \mu + C\sigma]$. We will use the same terminology for a Gaussian $N(\mu, \sigma^2)$.*

It will be used repeatedly that for a Gaussian (or Gaussian multiplier), $1 - \epsilon$-fraction of its mass is contained in its $O(\sqrt{\log 1/\epsilon})$-significant interval. We now state the main result of this section about approximating the indicator function of an interval as a weighted sum of Gaussian multipliers.

**Theorem B.3.** *Let $l$ be a positive real number and $0 < \epsilon < 0.1$ be a parameter. There is a function $f$ with the following properties*

1. *$f$ can be written a linear combination of Gaussian multipliers*

$$f(x) = w_1 M_{\mu_1, \sigma_1^2}(x) + \cdots + w_n M_{\mu_n, \sigma_n^2}(x)$$

*where $n = O((\log 1/\epsilon)^2)$ and $0 \leq w_1, \ldots, w_n \leq 1$*

2. *The $10\sqrt{\log 1/\epsilon}$-significant intervals of all of the $M_{\mu_i, \sigma_i^2}$ are contained in the interval $[-(1 + \epsilon)l, (1 + \epsilon)l]$*

3. *$0 \leq f(x) \leq 1 + \epsilon$ for all $x$*

4. *$1 - \epsilon \leq f(x) \leq 1 + \epsilon$ for all $x$ in the interval $[-l, l]$*

5. *$0 \leq f(x) \leq \epsilon$ for $x \geq (1 + \epsilon)l$ and $x \leq -(1 + \epsilon)l$*

## B.1 Approximating a Constant Function

Recall Lemma 2.5 (restated below) that allows us to approximate a constant function using an infinite sum of evenly spaced Gaussian multipliers.

**Lemma B.4** (Restated from Lemma 2.5). *Let $0 < \epsilon < 0.1$ be a parameter. Let $c$ be a real number such that $0 < c \leq (\log 1/\epsilon)^{-1/2}$. Define*

$$f(x) = \sum_{j=-\infty}^{\infty} \frac{c}{\sqrt{2\pi}} M_{cj\sigma,\sigma^2}(x).$$

*Then $1 - \epsilon \leq f(x) \leq 1 + \epsilon$ for all $x$.*

## B.2 Approximating an Interval

The next step in the proof of Theorem B.3 is to show how to approximate an interval using a finite number of Gaussian multipliers i.e. we need to show how to create the sharp transitions at the ends of the interval. In light of Lemma B.4, we can create a function satisfying the last four properties by taking $\widetilde{O}((1/\epsilon)^2)$ evenly spaced Gaussians multipliers with standard deviation $\epsilon^2 l$. However, this is too many components and we must reduce the number of components to $O(\log^2 1/\epsilon)$. The way we do this is by merging most of these components (all but the ones on the ends) into fewer components with larger standard deviation. We keep iterating this merging process and prove that we can eventually reduce the number of components to $O(\log^2 1/\epsilon)$.

First, the following result is an immediate consequence of Lemma B.4. It allows us to approximate a Gaussian with standard deviation $2\sigma$ as a weighted sum of Gaussians with standard deviation $\sigma$.

**Corollary B.5.** *Let $\epsilon$ be a parameter. Let $c$ be a real number such that $0 < c \leq 0.5(\log 1/\epsilon)^{-1/2}$. Let*

$$g(x) = \sum_{j=-\infty}^{\infty} \frac{\sqrt{2}c}{\sqrt{3\pi}} e^{-\frac{c^2 j^2}{6}} M_{cj\sigma,\sigma^2}(x).$$

*Then for all $x$,*

$$(1 - \epsilon)M_{0,4\sigma^2}(x) \leq g(x) \leq (1 + \epsilon)M_{0,4\sigma^2}(x).$$

*Proof.* Lemma B.4 (with $c \leftarrow \frac{2}{\sqrt{3}}c, \sigma \leftarrow \frac{2}{\sqrt{3}}\sigma$) implies that the function

$$f(x) = \sum_{j=-\infty}^{\infty} \frac{\sqrt{2}c}{\sqrt{3\pi}} M_{\frac{4}{3}cj\sigma, \frac{4}{3}\sigma^2}(x)$$

is between $1 - \epsilon$ and $1 + \epsilon$ everywhere. Now consider

$$f(x) \cdot M_{0,4\sigma^2}(x) = \sum_{j=-\infty}^{\infty} \frac{\sqrt{2}c}{\sqrt{3\pi}} M_{\frac{4}{3}cj\sigma, \frac{4}{3}\sigma^2}(x) M_{0,4\sigma^2}(x) = \sum_{j=-\infty}^{\infty} \frac{\sqrt{2}c}{\sqrt{3\pi}} e^{-\frac{x^2 + 3(x - \frac{4}{3}cj\sigma)^2}{8\sigma^2}}$$

$$= \sum_{j=-\infty}^{\infty} \frac{\sqrt{2}c}{\sqrt{3\pi}} e^{-\frac{c^2 j^2}{6}} e^{-\frac{(x - cj\sigma)^2}{2\sigma^2}} = \sum_{j=-\infty}^{\infty} \frac{\sqrt{2}c}{\sqrt{3\pi}} e^{-\frac{c^2 j^2}{6}} M_{cj\sigma,\sigma^2}(x).$$

∎

In the next lemma, we show when given a sum of evenly spaced Gaussians with standard deviation $\sigma$, we can replace almost all of them (except for ones on the ends) with a sum of fewer evenly spaced Gaussians with standard deviation $2\sigma$.

**Lemma B.6.** *Let $\epsilon$ be a parameter. Let $c$ be a real number such that $0 < c \leq 0.01(\log 1/\epsilon)^{-1/2}$. Let $b$ be a positive integer. Consider the function*

$$f(x) = \sum_{j=0}^{b} \frac{c}{\sqrt{2\pi}} M_{cj\sigma,\sigma^2}(x).$$

*Let $C = \lceil 10^2 c^{-1} \log(1/\epsilon)^{1/2} \rceil$. There is a function g of the form*

$$g(x) = \sum_{j=0}^{2C} \frac{w_j c}{\sqrt{2\pi}} M_{cj\sigma,\sigma^2}(x) + \sum_{j=b-2C}^{b} \frac{w_j c}{\sqrt{2\pi}} M_{cj\sigma,\sigma^2}(x) + \sum_{j=\lfloor C/2 \rfloor}^{\lceil (b-C)/2 \rceil} \frac{c}{\sqrt{2\pi}} M_{2cj\sigma,4\sigma^2}(x)$$

*where the $0 \le w_0, \ldots, w_{2C}, w_{b-2C}, \ldots, w_k \le 1$ are weights and*

$$\|f - g\|_\infty \le \epsilon^{10}.$$

*Proof.* Let $\epsilon' = \epsilon^{100}$. By Corollary B.5, for any real numbers $j, x$,

$$\left| M_{cj\sigma,4\sigma^2}(x) - \sum_{k=-\infty}^{\infty} \frac{\sqrt{2}c}{\sqrt{3\pi}} e^{-\frac{c^2 k^2}{6}} M_{c(k+j)\sigma,\sigma^2}(x) \right| \le \epsilon' M_{cj\sigma,4\sigma^2}(x).$$

Now we use the above inequality on each term of the last sum in the expression for $g(x)$.

$$\left| \sum_{j=\lfloor C/2 \rfloor}^{\lceil (b-C)/2 \rceil} \frac{c M_{2cj\sigma,4\sigma^2}(x)}{\sqrt{2\pi}} - \sum_{j=\lfloor C/2 \rfloor}^{\lceil (b-C)/2 \rceil} \sum_{k=-\infty}^{\infty} \frac{c^2}{\pi\sqrt{3}} e^{-\frac{c^2 k^2}{6}} M_{c(k+2j)\sigma,\sigma^2}(x) \right|$$

$$\le \epsilon' \sum_{j=\lfloor C/2 \rfloor}^{\lceil (b-C)/2 \rceil} \frac{c M_{2cj\sigma,4\sigma^2}(x)}{\sqrt{2\pi}} \le 2\epsilon'$$

where the last step follows from Lemma B.4. Now we rewrite the second sum in the LHS above. Let

$$S(x) = \sum_{j=\lfloor C/2 \rfloor}^{\lceil (b-C)/2 \rceil} \sum_{k=-\infty}^{\infty} \frac{c^2}{\pi\sqrt{3}} e^{-\frac{c^2 k^2}{6}} M_{c(k+2j)\sigma,\sigma^2}(x)$$

$$= \sum_{l=-\infty}^{\infty} \frac{c^2}{\pi\sqrt{3}} M_{cl\sigma,\sigma^2}(x) \sum_{j=\lfloor C/2 \rfloor}^{\lceil (b-C)/2 \rceil} e^{-\frac{c^2(l-2j)^2}{6}}.$$

Define

$$a_l = \sum_{j=\lfloor C/2 \rfloor}^{\lceil (b-C)/2 \rceil} e^{-\frac{c^2(l-2j)^2}{6}}.$$

First, by applying Lemma B.4 with parameters $c \leftarrow \frac{2}{\sqrt{3}}c, \sigma \leftarrow \sqrt{3}c^{-1}$, we have that for all real numbers $l$,

$$\left| \frac{\sqrt{3\pi}}{\sqrt{2}c} - \sum_{-\infty}^{\infty} e^{-\frac{c^2(l-2j)^2}{6}} \right| \le \frac{\epsilon'\sqrt{3\pi}}{\sqrt{2}c}.$$

By the way we chose $C$, we deduce that for all integers $l$ with $2C \le l \le b - 2C$,

$$\left| \frac{\sqrt{3\pi}}{\sqrt{2}c} - a_l \right| \le (2\epsilon') \frac{\sqrt{3\pi}}{\sqrt{2}c} \tag{2}$$

for all integers $0 \le l \le 2C$, or $b - 2C \le l \le b$,

$$a_l \le \frac{\sqrt{3\pi}}{\sqrt{2}c}(1 + 2\epsilon') \tag{3}$$

and finally for all integers $l < 0$ or $l > b$,

$$a_l \le \frac{\sqrt{3\pi}}{\sqrt{2}c}(2\epsilon'). \tag{4}$$

To obtain these inequalities, we simply use the fact that the terms in the sum

$$\sum_{-\infty}^{\infty} e^{-\frac{c^2(l-2j)^2}{6}}$$

decay exponentially when $j$ is far from $l/2$ so their total contribution is small.

Now we can set $w_0, \ldots, w_{2C}, w_{b-2C}, \ldots, w_k$ in the expresion for $g(x)$ as follows:

$$w_j = \max\left(0, 1 - \frac{\sqrt{2}c}{\sqrt{3}\pi} a_j\right).$$

It is clear that all of these weights are between 0 and 1. We now have that

$$\|f - g\|_\infty \leq 2\epsilon' + \left\| f(x) - \left( S(x) + \sum_{j=0}^{2C} \frac{w_j c}{\sqrt{2\pi}} M_{cj\sigma, \sigma^2}(x) + \sum_{j=b-2C}^{b} \frac{w_j c}{\sqrt{2\pi}} M_{cj\sigma, \sigma^2}(x) \right) \right\|_\infty$$

The expression inside the norm on the RHS can be rewritten as

$$\sum_{l=2C+1}^{b-2C-1} \left( \frac{c}{\sqrt{2\pi}} - \frac{c^2}{\pi\sqrt{3}} a_l \right) M_{cl\sigma, \sigma^2}(x) + \sum_{l=0}^{2C} \left( \frac{c}{\sqrt{2\pi}}(1 - w_l) - \frac{c^2}{\pi\sqrt{3}} a_l \right) M_{cl\sigma, \sigma^2}(x)$$

$$+ \sum_{l=b-2C}^{b} \left( \frac{c}{\sqrt{2\pi}}(1 - w_l) - \frac{c^2}{\pi\sqrt{3}} a_l \right) M_{cl\sigma, \sigma^2}(x) + \sum_{l=-\infty}^{-1} -\frac{c^2}{\pi\sqrt{3}} a_l M_{cl\sigma, \sigma^2}(x)$$

$$+ \sum_{l=b+1}^{\infty} -\frac{c^2}{\pi\sqrt{3}} a_l M_{cl\sigma, \sigma^2}(x)$$

and combining (2,3, 4), we deduce that the above has $L^\infty$ norm at most

$$\left\| (10\epsilon') \sum_{l=-\infty}^{\infty} \frac{c}{\sqrt{2\pi}} M_{cl\sigma, \sigma^2}(x) \right\|_\infty \leq 20\epsilon'.$$

where we used Lemma B.4. Thus, $\|f - g\|_\infty \leq 22\epsilon'$ and we are done. ∎

We can now prove Theorem B.3 by repeatedly applying Lemma B.6.

*Proof of Theorem B.3.* Let $c = 0.01(\log 1/\epsilon)^{-1/2}$. Let $K = \lceil \frac{1+0.5\epsilon}{c\epsilon^2} \rceil$

$$f_0(x) = \sum_{j=-K}^{K} \frac{c}{\sqrt{2\pi}} M_{cj\epsilon^2 l, \epsilon^4 l^2}(x).$$

Let $\epsilon' = \epsilon^{10}$. Using Lemma B.4, (and basic tail decay properties of a Gaussian) we get that

- $0 \leq f_0(x) \leq 1 + \epsilon'$ for all $x$
- $1 - \epsilon' \leq f_0(x) \leq 1 + \epsilon'$ for all $x$ in the interval $[-l, l]$
- $0 \leq f_0(x) \leq \epsilon'$ for $x \geq (1 + \epsilon)l$ and $x \leq -(1 + \epsilon)l$

Now we can apply Lemma B.6 to $f_0(x)$ to obtain

$$f_1(x) = \sum_{j=-K}^{-K+2C} \frac{w_j c}{\sqrt{2\pi}} M_{cj\epsilon^2 l, \epsilon^4 l^2}(x) + \sum_{j=K-2C}^{K} \frac{w_j c}{\sqrt{2\pi}} M_{cj\epsilon^2 l, \epsilon^4 l^2}(x)$$

$$+ \sum_{j=-\lceil (K-C)/2 \rceil}^{\lceil (K-C)/2 \rceil} \frac{c}{\sqrt{2\pi}} M_{2cj\epsilon^2 l, 4\epsilon^4 l^2}(x)$$

where $C = \lceil 10^2 c^{-1} \log(1/\epsilon)^{1/2} \rceil$, the $w_j$ are weights between 0 and 1, and

$$\|f_1 - f_0\| \leq \epsilon'.$$

Now we can apply Lemma B.6 again on the last sum in the expression for $f_1$. We have to do this at most $10 \log 1/\epsilon$ times before there are at most $O((\log 1/\epsilon)^2)$ components remaining. It is clear

that in this procedure, the $10\sqrt{\log 1/\epsilon}$-significant intervals of all of the Gaussian multipliers always remains in $[-(1+\epsilon)l, (1+\epsilon)l]$. Also, the total $L^\infty$ error incurred over all of the applications of Lemma B.6 is at most $10\epsilon' \log 1/\epsilon \leq \epsilon^9$. It is clear that all of the weights are always nonnegative and in the interval $[0,1]$. Thus, the final function $f$ satisfies

- $0 \leq f(x) \leq 1 + \epsilon$ for all $x$

- $1 - \epsilon \leq f(x) \leq 1 + \epsilon$ for all $x$ in the interval $[-l, l]$

- $0 \leq f(x) \leq \epsilon$ for $x \geq (1+\epsilon)l$ and $x \leq -(1+\epsilon)l$

and we are done. ∎

In light of Theorem B.3, we may make the following definition.

**Definition B.7.** *For parameters $\epsilon, l$, let $\mathcal{I}_{\epsilon,l}$ denote the function computed in Theorem B.3 for parameters $\epsilon, l/(1+\epsilon)$. We will also use $\mathcal{I}_{\epsilon,l}^{(a)}$ to denote the function $\mathcal{I}_{\epsilon,l}(x-a)$.*

**Remark.** *We define $\mathcal{I}_{\epsilon,l}$ as above because it will be convenient later to be able to say that the significant part of $\mathcal{I}_{\epsilon,l}$ is contained in the interval $[-l, l]$.*

## C   Nearly-Properly Learning GMMs: Full Version

In this section, we complete the proof of our main result for learning GMMs, Theorem 1.1. We localize the distribution by multiplying by a Gaussian multiplier $M_{\mu,\sigma^2}$. Note that the product of two Gaussians is still a Gaussian so multiplying a GMM by a Gaussian multiplier results in a re-weighted mixture of Gaussians. Roughly, we argue that the new weights on components of the mixture that are far away from the multiplier $M_{\mu,\sigma^2}$ are negligible so the resulting mixture is well-conditioned and we can then use Corollary A.5 to reconstruct the localized distribution. To reconstruct the entire distribution, we show that it suffices to sum together $\widetilde{O}(k)$ different localized reconstructions.

### C.1   Localizing with Gaussian Multipliers

Recall Claim 2.4 (restated below) which gives an explicit formula for what happens when we have a Gaussian $G_1 = N(\mu_1, \sigma_1^2)$ and we multiply it by a Gaussian multiplier $M_{\mu,\sigma^2}(x)$.

**Claim C.1** (Restated from Claim 2.4). *We have the identity*

$$M_{\mu,\sigma^2}(x)N(\mu_1, \sigma_1^2) = \frac{1}{\sqrt{1 + \frac{\sigma_1^2}{\sigma^2}}} e^{-\frac{(\mu_1 - \mu)^2}{2(\sigma_1^2 + \sigma^2)}} N\left(\frac{\mu\sigma_1^2 + \mu_1\sigma^2}{\sigma_1^2 + \sigma^2}, \frac{\sigma_1^2\sigma^2}{\sigma_1^2 + \sigma^2}\right).$$

### C.2   Building Blocks

We first consider reconstructing a GMM $\mathcal{M} = w_1 G_1 + \dots w_k G_k$ after multiplying by a Gaussian multiplier $M_{\mu,\sigma^2}$. As a corollary of Claim C.1, we know that when the $C$-significant intervals (recall Definition B.2) of a Gaussian $G_j$ and the multiplier $M_{\mu,\sigma^2}(x)$ are disjoint for large $C$, then the $L^1$ norm of their product is $e^{-\Omega(C^2)}$. In particular this means that after multiplying by $M_{\mu,\sigma^2}$, the only components that remain relevant are those that have nontrivial overlap with the multiplier $M_{\mu,\sigma^2}$. The only way these components will not form a well-conditioned mixture is if there is some $G_j$ that is very thin (i.e. $\sigma_j << \sigma$) and overlaps with $M_{\mu,\sigma^2}$. As long as this doesn't happen, we can apply Corollary A.5. We formalize this below.

**Corollary C.2.** *Let $\mathcal{M} = w_1 G_1 + \dots + w_k G_k$ be an arbitrary mixture of Gaussians where $G_i = N(\mu_i, \sigma_i^2)$. Let $\epsilon > 0$ be some parameter and let $l = \lceil \sqrt{\log(1/\epsilon)} \rceil$. Assume we are given access to a distribution $f$. Let $M_{\mu,\sigma^2}$ be a Gaussian multiplier. Assume that for all $i \in [k]$, either $\sigma_i \geq 4l\sigma$ or the $10l$-significant intervals of $G_i$ and $M_{\mu,\sigma^2}$ do not intersect. Then in $\mathrm{poly}(1/\epsilon)$ time and with high probability, we can compute a weighted sum $\widetilde{M}$ of at most $O(\log(1/\epsilon))$ Gaussians such that*

$$\|\widetilde{M} - M_{\mu,\sigma^2} f\|_1 \leq \epsilon + \mathrm{poly}(\log(1/\epsilon))\|M_{\mu,\sigma^2}(\mathcal{M} - f)\|_1.$$

*Proof.* We compute $M_{\mu,\sigma^2} f$ and let $C = \|M_{\mu,\sigma^2} f\|_1$. If $C \leq \epsilon$ then we may simple output $0$. Otherwise, we will apply Corollary A.5 on $M_{\mu,\sigma^2} f / C$ and multiply the result by $C$. We must first verify the conditions of Corollary A.5. Let $S \subset [k]$ be the indices such that the $10l$-significant intervals of $G_i$ and $M_{\mu,\sigma^2}$ intersect. First for $i \notin S$, by Claim C.1,

$$\|G_i M_{\mu,\sigma^2}\|_1 \leq e^{-\frac{(\mu_i - \mu)^2}{2(\sigma_i^2 + \sigma^2)}} \leq e^{-10l^2} \leq \epsilon^{10} .$$

Let

$$\mathcal{M}' = \sum_{i \in S} w_i G_i .$$

Then we know

$$\|\frac{M_{\mu,\sigma^2} f}{C} - \frac{M_{\mu,\sigma^2} \mathcal{M}'}{C}\|_1 \leq \epsilon^9 + \frac{\|M_{\mu,\sigma^2}(f - \mathcal{M})\|_1}{C}$$

Next, for $i \in S$,

$$G_i M_{\mu,\sigma^2} = w_i' N\left(\frac{\mu \sigma_i^2 + \mu_i \sigma^2}{\sigma_i^2 + \sigma^2}, \frac{\sigma_i^2 \sigma^2}{\sigma_i^2 + \sigma^2}\right)$$

for some weight $w_i'$ and since we must have $\sigma_i \geq 4l\sigma$, then

$$\frac{\sigma^2}{2} \leq \frac{\sigma_i^2 \sigma^2}{\sigma_i^2 + \sigma^2} \leq \sigma^2$$

$$\left|\frac{\mu \sigma_i^2 + \mu_i \sigma^2}{\sigma_i^2 + \sigma^2} - \mu\right| = \left|\frac{(\mu_i - \mu)\sigma^2}{\sigma_i^2 + \sigma^2}\right| \leq \frac{l(\sigma + \sigma_i)\sigma^2}{\sigma_i^2 + \sigma^2} \leq \frac{\sigma}{2}$$

Let

$$\mathcal{M}'' = \sum_{i \in S} \frac{w_i'}{\sum_{i \in S} w_i'} N\left(\frac{\mu \sigma_i^2 + \mu_i \sigma^2}{\sigma_i^2 + \sigma^2}, \frac{\sigma_i^2 \sigma^2}{\sigma_i^2 + \sigma^2}\right) .$$

Then we deduce, since $\|M_{\mu,\sigma^2} f / C\|_1 = 1$, that

$$\|\frac{M_{\mu,\sigma^2} f}{C} - \mathcal{M}''\|_1 \leq \|\mathcal{M}'' - \frac{M_{\mu,\sigma^2} \mathcal{M}'}{C}\|_1 + \|\frac{M_{\mu,\sigma^2} f}{C} - \frac{M_{\mu,\sigma^2} \mathcal{M}'}{C}\|_1$$

$$\leq 2\|\frac{M_{\mu,\sigma^2} f}{C} - \frac{M_{\mu,\sigma^2} \mathcal{M}'}{C}\|_1$$

$$\leq \epsilon^8 + 2\frac{\|M_{\mu,\sigma^2}(f - \mathcal{M})\|_1}{C}$$

and further, after applying a suitable linear transformation (taking $(\mu, \sigma^2) \to (0, 1)$) that the mixture $\mathcal{M}''$ is 0.5-well-conditioned. Thus, we can apply Corollary A.5 and compute a weighted sum of $O(\log(1/\epsilon))$ Gaussians, $\widetilde{\mathcal{M}}$ such that

$$\|\frac{M_{\mu,\sigma^2} f}{C} - \widetilde{\mathcal{M}}\|_1 \leq \text{poly}(\log(1/\epsilon))\left(\epsilon^8 + \frac{\|M_{\mu,\sigma^2}(f - \mathcal{M})\|_1}{C}\right) .$$

Now we can simply output $C\widetilde{\mathcal{M}}$ (which is still a weighted sum of $O(\log(1/\epsilon))$ Gaussians) and we are done. ∎

Recall that Theorem B.3 shows how to express an interval as a sum of Gaussian multipliers. Combining Theorem B.3 with Corollary C.2, we show that we can approximate a GMM over an interval as long as the interval does not overlap with a component that is much thinner than it.

**Lemma C.3.** *Let $\mathcal{M} = w_1 G_1 + \cdots + w_k G_k$ be an arbitrary mixture of Gaussians where $G_i = N(\mu_i, \sigma_i^2)$. Let $\epsilon > 0$ be some parameter and let $l = \lceil\sqrt{\log(1/\epsilon)}\rceil$. Assume we are given access to a distribution $f$. Let $I = [a, b]$ be an interval. Assume that for all $i \in [k]$, either $\sigma_i \geq (b - a)$ or the $10l$-significant interval of $G_i$ does not intersect $I$. Then in $\text{poly}(1/\epsilon)$ time and with high probability, we can compute a weighted sum $\widetilde{\mathcal{M}}$ of at most $\text{poly}(\log(1/\epsilon))$ Gaussians such that*

$$\|\widetilde{\mathcal{M}} - f \cdot 1_I\|_1 \leq \text{poly}(\log(1/\epsilon))\left(\epsilon + \|1_I(\mathcal{M} - f)\|_1\right)$$

*where $1_I$ denotes the indicator function of $I$.*

*Proof.* Consider the function $\mathcal{I} = \mathcal{I}_{\epsilon,(b-a)/2}^{(a+b)/2}$ (recall Definition B.7). Now note that by Theorem B.3, $\mathcal{I}$ can be written in the form

$$\mathcal{I} = \widetilde{w_1} M_{\widetilde{\mu_1},\widetilde{\sigma_1}^2} + \cdots + \widetilde{w_n} M_{\widetilde{\mu_n},\widetilde{\sigma_n}^2}$$

where $n = O(\log^2 1/\epsilon)$. Furthermore, for all $i \in [n]$, we have $0 \leq \widetilde{w_i} \leq 1$ and $\widetilde{\sigma_i} \leq (b-a)/(4l)$ and the $10l$-significant intervals of $M_{\widetilde{\mu_i},\widetilde{\sigma_i}^2}$ are all contained in the interval $[a,b]$. Thus we can apply Corollary C.2 on $M_{\widetilde{\mu_i},\widetilde{\sigma_i}^2} f$ for all $i \in [n]$. Adding the results with the corresponding weights $\widetilde{w_1}, \ldots, \widetilde{w_n}$, we obtain a function $\widetilde{\mathcal{M}}$ that is a weighted sum of at most $\mathrm{poly}(\log(1/\epsilon))$ Gaussians such that

$$\|\widetilde{\mathcal{M}} - f\mathcal{I}\|_1 \leq \mathrm{poly}(\log(1/\epsilon)) \left( \epsilon + \sum_{i=1}^{n} \widetilde{w_i} \|M_{\widetilde{\mu_i},\widetilde{\sigma_i}^2}(\mathcal{M}-f)\|_1 \right)$$

$$= \mathrm{poly}(\log(1/\epsilon)) \left( \epsilon + \|\mathcal{I}(\mathcal{M}-f)\|_1 \right).$$

Thus,

$$\|\widetilde{\mathcal{M}} - \mathcal{M}\mathcal{I}\|_1 \leq \mathrm{poly}(\log(1/\epsilon)) \left( \epsilon + \|\mathcal{I}(\mathcal{M}-f)\|_1 \right). \tag{5}$$

Next, by the properties in Theorem B.3,

$$\|\mathcal{M}(\mathcal{I} - 1_I)\|_1 \leq \epsilon \int_{-\infty}^{\infty} \mathcal{M} + \int_{a}^{a+\epsilon(b-a)} \mathcal{M} + \int_{b-\epsilon(b-a)}^{b} \mathcal{M}$$

$$\leq \epsilon + \left( \sum_{j=1}^{k} w_j \int_{a}^{a+\epsilon(b-a)} G_j + w_j \int_{b-\epsilon(b-a)}^{b} G_j \right).$$

Consider one of the component Gaussians $G_j$ where $j \in [k]$. If the $10l$-significant interval of $G_j$ does not intersect $[a,b]$ then it is clear that the total mass of $G_j$ on the interval $[a,b]$ is at most $\epsilon$. Otherwise, we know that the standard deviation of $G_j$ is at least $b-a$ which means that its mass on the set $[a, a+\epsilon(b-a)] \cup [b-\epsilon(b-a), b]$ is at most $O(\epsilon)$. Thus we conclude that

$$\|\mathcal{M}(\mathcal{I} - 1_I)\|_1 \leq O(\epsilon). \tag{6}$$

Also note that by the properties in Theorem B.3

$$\|\mathcal{I}(\mathcal{M}-f)\|_1 \leq \epsilon \int_{-\infty}^{\infty} |\mathcal{M}-f| + 2\int_{a}^{b} |\mathcal{M}-f| \leq 2(\epsilon + \|1_I(\mathcal{M}-f)\|_1). \tag{7}$$

Putting together (5, 6, 7), we conclude

$$\|\widetilde{\mathcal{M}} - f \cdot 1_I\|_1 \leq \|1_I(\mathcal{M}-f)\|_1 + \|\widetilde{\mathcal{M}} - \mathcal{M}1_I\|_1 \leq \mathrm{poly}(\log(1/\epsilon)) \left( \epsilon + \|1_I(\mathcal{M}-f)\|_1 \right)$$

and we are done. $\blacksquare$

### C.3 Structural Properties

Lemma C.3 allows us to reconstruct the unknown GMM $\mathcal{M}$ over certain intervals. However, it cannot be applied to an arbitrary interval (because an interval may overlap with a component that is too thin). We will now prove several structural results that will imply that there exist $\widetilde{O}(k)$ intervals for which the conditions of Lemma C.3 are satisfied (i.e. these intervals do not overlap with components that are much thinner than themselves) and such that the union of these intervals contains most of the mass of $\mathcal{M}$. Then, to complete the proof of Theorem 1.1, we show how to find such a set of $\widetilde{O}(k)$ intervals using a dynamic program.

First, we define a modified density function for a GMM $\mathcal{M} = w_1 G_1 + \cdots + w_k G_k$ where we modify each component Gaussian by restricting it to its $10l$-significant interval (and making it 0 outside). It is clear that this modified function is close to $\mathcal{M}$ in $L^1$-distance but it will be convenient to use in the analysis later on.

**Definition C.4.** *For a mixture of Gaussians $\mathcal{M} = w_1 G_1 + \cdots + w_k G_k$ where $G_j = N(\mu_j, \sigma_j^2)$ and a parameter $l$, define the function $\mathcal{M}_{sig,l}(x)$ to be, at each point $x \in \mathbb{R}$, equal to the weighted sum of the components $G_j$ of $\mathcal{M}$ such that $x$ is in the $10l$-significant interval of $G_j$. Formally,*

$$\mathcal{M}_{sig,l}(x) = \sum_{\substack{j \text{ such that} \\ |x-\mu_j| \leq 10l\sigma_j}} w_j G_j(x).$$

The following claim is immediate from the definition.

**Claim C.5.** *Let $\epsilon > 0$ be some parameter and let $l = \lceil \sqrt{\log 1/\epsilon} \rceil$. Then*

$$\|\mathcal{M} - \mathcal{M}_{sig,l}\|_1 \le \epsilon$$

*Proof.* The inequality holds because the total mass of a Gaussian outside of its $10l$-significant interval is at most $\epsilon$. ■

We now present our first structural result.

**Claim C.6.** *Let $\mathcal{M} = w_1 G_1 + \cdots + w_k G_k$ be an arbitrary mixture of Gaussians where $G_i = N(\mu_i, \sigma_i^2)$. Let $\epsilon > 0$ be some parameter and let $l = \lceil \sqrt{\log(1/\epsilon)} \rceil$. There exist disjoint intervals $I_1, \dots, I_n$ with lengths, say $t_1, \dots, t_n$, where $n \le 50kl$ with the following property:*

- *For each interval $I_i$, for all $j \in [k]$ either the the $10l$-significant interval of $G_j$ is disjoint from $I_i$ or $\sigma_j \ge t_i$*

- *We have*
$$\|\mathcal{M} - (1_{I_1} + \cdots + 1_{I_n})\mathcal{M}\|_1 \le \epsilon$$

*Proof.* Sort the Gaussians by their standard deviations, WLOG $\sigma_1 \le \cdots \le \sigma_k$. Now we will create several intervals $A_1, A_2, \dots$ and we will also associate each interval with one of the Gaussians $G_1, \dots, G_k$ which we will call its parent.

First, set $A_1$ to be the $10l$-significant interval of $G_1$. Next, we will process the Gaussians $G_2, \dots, G_k$ in order. For $G_j$, assume that the intervals we have created so far are $A_1, \dots, A_m$ (which will be disjoint by construction). Now consider the $10l$-significant interval of $G_j$, say $L_j$. Note that removing the union of the intervals $A_1, \dots, A_m$ from $L_j$ divides $L_j$ into several (at most $m + 1$) disjoint intervals. We label these intervals $A_{m+1}, A_{m+2}, \dots$ and set all of their parents to be $G_j$. We then move onto $G_{j+1}$ and repeat the above process. The following properties are immediate from the construction:

1. If the parent of $A_i$ is $G_j$ then the length of $A_i$ is at most $20l\sigma_j$

2. The union of all of the $A_i$ whose parent is among $G_1, \dots, G_j$ contains the $10l$-significant intervals of all of $G_1, \dots, G_j$

3. If the parent of $A_i$ is $G_j$, then $A_i$ is disjoint from the $10l$-significant intervals of $G_1, \dots, G_{j-1}$

Now we claim that at the end of the algorithm, the total number of intervals is at most $2k$. To see this, say that after processing $G_{j-1}$, the intervals we have created are $A_1, \dots A_m$. Now consider the potential that is the number of intervals $m$ plus the number of connected components in $A_1 \cup \cdots \cup A_m$. This potential can increase by at most $2$ when processing $G_j$ so thus the total number of intervals at the end of the execution is at most $2k$. We will now assume that the intervals at the end of the execution are $A_1, \dots, A_{2k}$ (if there are less than $2k$ intervals then add a bunch of dummy intervals of length 0).

We now describe a post-processing step. For each of $A_1, \dots, A_{2k}$, if its parent is $G_j$ then divide it into intervals of length at most $\sigma_j$ and assign $G_j$ as the parent of all of these intervals. By property 1, we can ensure that this creates a total of at most $50kl$ intervals, say $I_1, \dots, I_n$ where $n \le 50kl$. We use $t_1, \dots, t_n$ to denote their lengths. We now prove that this set of intervals satisfies the desired properties. First, note that the following properties are immediate from the construction:

1. If the parent of $I_i$ is $G_j$ then $I_i$ is contained in the $10l$-significant interval of $G_j$ and $t_i \le \sigma_j$

2. The union of all of $I_1, \dots, I_n$ contains the $10l$-significant intervals of all of $G_1, \dots, G_k$

3. If the parent of $I_i$ is $G_j$, then $I_i$ is disjoint from the $10l$-significant intervals of $G_1, \dots, G_{j-1}$

The first of the desired properties is clear since by construction if the parent of $I_i$ is $G_j$, then $t_i \leq \sigma_j$ and it must be disjoint from the $10l$-significant intervals of all of $G_1, \ldots, G_{j-1}$ (where recall $G_1, \ldots, G_k$ are sorted in increasing order of their standard deviation). Now it remains to verify the second property. Consider the function $\mathcal{M}_{\text{sig},l}(x)$. Recall by Claim C.5,

$$\|\mathcal{M} - \mathcal{M}_{\text{sig},l}\|_1 \leq \epsilon \tag{8}$$

Next observe that

$$\mathcal{M}_{\text{sig},l} = (1_{I_1} + \cdots + 1_{I_n})\mathcal{M}_{\text{sig},l}.$$

Combining the above, we have

$$\|\mathcal{M} - (1_{I_1} + \cdots + 1_{I_n})\mathcal{M}_{\text{sig},l}\|_1 \leq \epsilon.$$

However, note that we have

$$(1_{I_1} + \cdots + 1_{I_n})\mathcal{M}_{\text{sig},l} \leq (1_{I_1} + \cdots + 1_{I_n})\mathcal{M} \leq \mathcal{M}$$

everywhere along the real line. Thus, we immediately get the desired inequality. ∎

The next structural result shows that the intervals $I_1, \ldots, I_n$ obtained in Claim C.6 are "findable" in the sense that if we draw many samples from $\mathcal{M}$ (or from a distribution $f$ that is close to $\mathcal{M}$), then with high probability, there will be samples close to the endpoints of each of $I_1, \ldots, I_n$. This will mean that algorithmically, it suffices to draw sufficiently many samples and then only consider intervals whose endpoints are given by a pair of samples.

**Claim C.7.** *Let $\mathcal{M} = w_1 G_1 + \cdots + w_k G_k$ be an arbitrary mixture of Gaussians where $G_i = N(\mu_i, \sigma_i^2)$. Let $\epsilon > 0$ be some parameter and let $l = \lceil \sqrt{\log(1/\epsilon)} \rceil$. Let $f$ be a distribution. Assume we are given samples from $f$, say $x_1, \ldots, x_Q$ for some sufficiently large $Q = \text{poly}(k/\epsilon)$. Then with high probability, there exists pairs $\{x_{a_1}, x_{b_1}\}, \ldots, \{x_{a_n}, x_{b_n}\}$ such that*

- *The intervals $J_1 = [x_{a_1}, x_{b_1}], \ldots, J_n = [x_{a_n}, x_{b_n}]$ are disjoint*

- *$n \leq 50kl$*

- *For each interval $J_i$, for all $j \in [k]$ either the the $10l$-significant interval of $G_j$ is disjoint from $J_i$ or $\sigma_j \geq |x_{b_i} - x_{a_i}|$*

- 
$$\|\mathcal{M} - (1_{J_1} + \cdots + 1_{J_n})\mathcal{M}\|_1 \leq 4(\epsilon + \|\mathcal{M} - f\|_1).$$

*Proof.* Let $I_1, \ldots, I_n$ be the intervals computed in Claim C.6 applied to the mixture $\mathcal{M}$ and assume that their lengths are $t_1, \ldots, t_n$. Let $C = \lceil (k/\epsilon)^2 \rceil$. For each interval $I_i$, divide it into $C$ subintervals $I_i^1, \ldots, I_i^C$ of length $t_i/C$ and assume that these subintervals are sorted in order. We say one of these subintervals is good if

$$\int_{I_i^j} f \geq (\epsilon/k)^{10}.$$

For an index $i$, let $c_i, d_i$ be the smallest and largest index such that $I_i^{c_i}, I_i^{d_i}$ are good respectively. Then with high probability for all $i$, there will be samples, say $x_{a_i}, x_{b_i}$ in $I_i^{c_i}$ and $I_i^{d_i}$. Now we will form the intervals $J_i = [x_{a_i}, x_{b_i}]$. The first two of the desired properties are clear. The third follows from the statement of Claim C.6. It remains to verify the last. Similar to the proof of Claim C.6, we consider the function $\mathcal{M}_{\text{sig},l}$. Note that

$$\int_{I_i \setminus J_i} \mathcal{M}_{\text{sig},l} \leq \int_{I_i^1} \mathcal{M}_{\text{sig},l} + \cdots + \int_{I_i^{c_i}} \mathcal{M}_{\text{sig},l} + \int_{I_i^{d_i}} \mathcal{M}_{\text{sig},l} + \cdots + \int_{I_i^C} \mathcal{M}_{\text{sig},l}$$

$$\leq \|1_{I_i}(\mathcal{M}_{\text{sig},l} - f)\|_1 + \int_{I_i^1} f + \cdots + \int_{I_i^{c_i-1}} f + \int_{I_i^{d_i+1}} f + \cdots + \int_{I_i^C} f$$

$$+ \int_{I_i^{c_i}} \mathcal{M}_{\text{sig},l} + \int_{I_i^{d_i}} \mathcal{M}_{\text{sig},l}$$

$$\leq \|1_{I_i}(\mathcal{M}_{\text{sig},l} - f)\|_1 + (\epsilon/k)^2 + \int_{I_i^{c_i}} \mathcal{M}_{\text{sig},l} + \int_{I_i^{d_i}} \mathcal{M}_{\text{sig},l}$$

where the last step follows by the minimality and maximality of $c_i, d_i$. Now by construction, the only Gaussians among $G_1, \ldots, G_k$ whose $10l$-significant intervals intersect $I_i$ must have standard deviation at least $t_i$. Since $I_i^{c_i}, I_i^{d_i}$ each have length $(\epsilon/k)^2 t_i$, we conclude

$$\int_{I_i \setminus J_i} \mathcal{M}_{\text{sig},l} \leq 3(\epsilon/k)^2 + \|1_{I_i}(\mathcal{M}_{\text{sig},l} - f)\|_1 \,.$$

Thus, we have

$$\|(1_{I_1} + \cdots + 1_{I_n})\mathcal{M}_{\text{sig},l} - (1_{J_1} + \cdots + 1_{J_n})\mathcal{M}_{\text{sig},l}\|_1 \leq \epsilon + \|(\mathcal{M}_{\text{sig},l} - f)\|_1 \leq 2\epsilon + \|(\mathcal{M} - f)\|_1 \,.$$

where we used Claim C.5 in the last step. Now, using the statement of Claim C.6, we deduce

$$\|\mathcal{M} - (1_{J_1} + \cdots + 1_{J_n})\mathcal{M}_{\text{sig},l}\|_1 \leq 4\epsilon + \|(\mathcal{M} - f)\|_1 \,.$$

Since

$$(1_{J_1} + \cdots + 1_{J_n})\mathcal{M}_{\text{sig},l} \leq (1_{J_1} + \cdots + 1_{J_n})\mathcal{M} \leq \mathcal{M}$$

everywhere along the real line, we immediately get the desired inequality. ∎

## C.4 Finishing the Proof

We can now prove the key lemma and then Theorem 1.1 will follow as an immediate consequence since we can use the improper learner in Corollary 2.10. The lemma states that given explicit access to a distribution $f$ that is $\epsilon$-close to a GMM, $\mathcal{M}$, with $k$ components, we can output a GMM, $\widetilde{\mathcal{M}}$, with $\widetilde{O}(k)$ components that is $\widetilde{O}(\epsilon)$-close to $f$. At a high-level, the proof involves attempting to reconstruct $f$ over various intervals using Lemma C.3 and then using a dynamic program to find a union of $\widetilde{O}(k)$ such intervals that approximates the entire function. We use Claim C.7 to argue that such a solution exists so our dynamic program must find it.

**Lemma C.8.** *Let $\mathcal{M} = w_1 G_1 + \cdots + w_k G_k$ be an arbitrary mixture of Gaussians where $G_i = N(\mu_i, \sigma_i^2)$. Let $\epsilon > 0$ be some parameter. Assume we are given access to a distribution $f$. Then in* $\text{poly}(k/\epsilon)$ *time and with high probability, we can compute a mixture $\widetilde{\mathcal{M}}$ of at most $k\text{poly}(\log(k/\epsilon))$ Gaussians such that*

$$\|\widetilde{\mathcal{M}} - f\|_1 \leq \text{poly}(\log(k/\epsilon)) \left(\epsilon + \|\mathcal{M} - f\|_1\right) \,.$$

*Proof.* Note that it suffices to compute a weighted sum of Gaussians that satisfies the desired inequality since rescaling such a weighted sum to a mixture will at most increase the $L^1$ error by a factor of 2. Thus, from now on, we will not worry about ensuring the mixing weights sum to 1.

Let $\gamma = \epsilon/k$ and $l = \lceil \sqrt{\log 1/\gamma} \rceil$. First draw $Q = \text{poly}(k/\epsilon)$ samples $x_1, \ldots, x_Q$ from $f$ for sufficiently large $Q$ that we can apply Claim C.7 with $\epsilon \leftarrow \gamma$. While we do not know what the intervals $J_1, \ldots, J_n$ are, we will set up a dynamic program to find a set of at most $50kl$ intervals that we can reconstruct $f$ over each one using Lemma C.3 and such that these intervals contain essentially all of the mass of $f$.

For each pair $x_a, x_b$ with $a, b \in \{1, 2, \ldots, Q\}$, apply Lemma C.3 with parameter $\epsilon \leftarrow \gamma$ to attempt to approximate $f$ on the interval $[x_a, x_b]$. Let the output obtained be $\widetilde{\mathcal{M}}_{x_a, x_b}$. Note that sometimes the algorithm will fail to output a good approximation to $f$ (because the assumptions of Lemma C.3 fail) but we can ensure that the output is a weighted sum of at most $\text{poly}(\log 1/\gamma)$ Gaussians. In the proof we will only use the fact that when the assumptions of Lemma C.3 hold, then our approximation to $f$ restricted to the interval will be accurate.

Now we show how to set up the dynamic program. WLOG the points $x_1, \ldots, x_Q$ are sorted in nondecreasing order. We also use the convention that $x_0 = -\infty, x_{Q+1} = \infty$. Now, we maintain the following state for each index $0 \leq j \leq Q+1$, and integer $c \leq 50kl$: the best approximation to $f$ from $(-\infty, x_j]$ using a sum of $\widetilde{\mathcal{M}}_{x_a, x_b}$ over at most $c$ intervals. Formally,

**Dynamic Program:** Let $DP_{j,c}$ be the minimum over all sets $S$ of $c$ pairs $(a^{(1)}, b^{(1)}), \ldots, (a^{(c)}, b^{(c)}) \in [j] \times [j]$ such that $a^{(1)} < b^{(1)} \leq a^{(2)} < \cdots < b^{(c)}$ of

$$\|f \cdot 1_{(-\infty, x_j]} - \left(1_{[x_{a^{(1)}}, x_{b^{(1)}}]}\widetilde{\mathcal{M}}_{x_{a^{(1)}}, x_{b^{(1)}}} + \cdots + 1_{[x_{a^{(c)}}, x_{b^{(c)}}]}\widetilde{\mathcal{M}}_{x_{a^{(c)}}, x_{b^{(c)}}}\right)\|_1$$
$$+ \|\widetilde{\mathcal{M}}_{x_{a^{(1)}}, x_{b^{(1)}}} - 1_{[x_{a^{(1)}}, x_{b^{(1)}}]}\widetilde{\mathcal{M}}_{x_{a^{(1)}}, x_{b^{(1)}}}\|_1 + \cdots + \|\widetilde{\mathcal{M}}_{x_{a^{(c)}}, x_{b^{(c)}}} - 1_{[x_{a^{(c)}}, x_{b^{(c)}}]}\widetilde{\mathcal{M}}_{x_{a^{(c)}}, x_{b^{(c)}}}\|_1 \,.$$

Note that the first term represents the approximation error compared to $f$. We must truncate each function $\widetilde{\mathcal{M}}_{x_{a^{(1)}}, x_{b^{(1)}}}$ to its corresponding interval $[x_{a^{(1)}}, x_{b^{(1)}}]$ in order for the problem to be solvable via dynamic programming because otherwise previous choices would affect later ones. Thus, we also need to add the additonal terms that represent the error from truncation. Note that the $L^1$ distance can be estimated using Claim 2.11. We can solve the dynamic program in polynomial time because from each state $DP_{j,c}$, we simply consider adding all possible intervals among $x_j, x_{j+1}, \ldots, x_Q$ as the next one.

Now we prove that there is a good solution to the dynamic program for which the objective (for $j = Q + 1$) is small. Let $a_1, b_1, \ldots, a_n, b_n$ be the indices obtained in Claim C.7. We claim that setting
$$(a^{(1)}, b^{(1)}) = (a_1, b_1), \ldots, (a^{(n)}, b^{(n)}) = (a_n, b_n)$$
results in the objective function being small. Let $J_i = [a_i, b_i]$ for all $i$. Let
$$\widetilde{\mathcal{M}}^{\text{good}} = 1_{[x_{a_1}, x_{b_1}]} \widetilde{\mathcal{M}}_{x_{a_1}, x_{b_1}} + \cdots + 1_{[x_{a_n}, x_{b_n}]} \widetilde{\mathcal{M}}_{x_{a_n}, x_{b_n}}.$$

The guarantee from Lemma C.3 implies that for all $i$,
$$\|\widetilde{\mathcal{M}}_{x_{a_i}, x_{b_i}} - 1_{[x_{a_i}, x_{b_i}]} \widetilde{\mathcal{M}}_{x_{a_i}, x_{b_i}}\|_1 \leq \text{poly}(\log 1/\gamma)(\gamma + \|1_{J_i}(\mathcal{M} - f)\|_1) \tag{9}$$

and thus, using the guarantee from Lemma C.3 again, we have
$$\|\widetilde{\mathcal{M}}^{\text{good}} - (1_{J_1} + \cdots + 1_{J_n})f\| \leq \|\widetilde{\mathcal{M}}_{x_{a_1}, x_{b_1}} - 1_{J_1} \cdot f\|_1 + \cdots + \|\widetilde{\mathcal{M}}_{x_{a_n}, x_{b_n}} - 1_{J_n} \cdot f\|_1$$
$$+ \|\widetilde{\mathcal{M}}_{x_{a_1}, x_{b_1}} - 1_{[x_{a_1}, x_{b_1}]} \widetilde{\mathcal{M}}_{x_{a_1}, x_{b_1}}\|_1 + \cdots + \|\widetilde{\mathcal{M}}_{x_{a_n}, x_{b_n}} - 1_{[x_{a_n}, x_{b_n}]} \widetilde{\mathcal{M}}_{x_{a_n}, x_{b_n}}\|_1$$
$$\leq n\gamma \text{poly}(\log 1/\gamma) + \text{poly}(\log 1/\gamma) \left( \|1_{J_1}(\mathcal{M} - f)\|_1 + \cdots + \|1_{J_n}(\mathcal{M} - f)\|_1 \right)$$
$$\leq \text{poly}(\log 1/\gamma) \left( \epsilon + \|\mathcal{M} - f\|_1 \right).$$

However also recall that by Claim C.7,
$$\|\mathcal{M} - (1_{J_1} + \cdots + 1_{J_n})\mathcal{M}\|_1 \leq 4(\gamma + \|\mathcal{M} - f\|_1).$$

Combining the above two inequalities, we get
$$\|\widetilde{\mathcal{M}}^{\text{good}} - f\|_1 \leq \text{poly}(\log 1/\gamma) \left( \epsilon + \|\mathcal{M} - f\|_1 \right).$$

Finally, combining the above with (9) implies that the objective value of the dynamic program is at most $\text{poly}(\log 1/\gamma) \left( \epsilon + \|\mathcal{M} - f\|_1 \right)$. Finally it remains to note that the objective of the dynamic program (for $j = Q + 1$) is an upper bound on
$$\left\| f - \left( \widetilde{\mathcal{M}}_{x_{a^{(1)}}, x_{b^{(1)}}} + \cdots + \widetilde{\mathcal{M}}_{x_{a^{(c)}}, x_{b^{(c)}}} \right) \right\|_1$$

so thus, we can simply output the solution $\widetilde{\mathcal{M}} = \widetilde{\mathcal{M}}_{x_{a^{(1)}}, x_{b^{(1)}}} + \cdots + \widetilde{\mathcal{M}}_{x_{a^{(c)}}, x_{b^{(c)}}}$ and we are guaranteed to have
$$\|f - \widetilde{\mathcal{M}}\|_1 \leq \text{poly}(\log 1/\gamma) \left( \epsilon + \|\mathcal{M} - f\|_1 \right).$$

It is clear that $\widetilde{\mathcal{M}}$ is a weighted sum of at most $k\text{poly}(\log 1/\gamma)$ Gaussians (since for each interval there are $\text{poly}(\log 1/\gamma)$ Gaussians and there are at most $50kl$ total intervals). It is clear that all of the steps run in $\text{poly}(k/\epsilon)$ time and we are done. ∎

Now we can complete the proof of our main theorem, Theorem 1.1.

*Proof of Theorem 1.1.* We can apply Corollary 2.10 to learn a distribution $f$ such that $d_{\text{TV}}(\mathcal{M}, f) \leq O(\epsilon)$. We can then apply Lemma C.8 using $f$. Note that since $f$ is a piecewise polynomial, we can perform all of the explicit computations with the density function that are used in the proof of Lemma C.8. It is immediate that the output of Lemma C.8 must satisfy
$$d_{\text{TV}}(\widetilde{\mathcal{M}}, \mathcal{M}) \leq \widetilde{O}(\epsilon)$$

so we are done. ∎

# D  Well-Conditioned Case: Sparse Fourier Reconstruction

We now move onto our results on sparse Fourier reconstruction. As with GMMs, we will first consider the well-conditioned case. Here, this means a function that has its Fourier support contained in one interval that is not too long i.e., all of its Fourier mass is not too spread out. Note that WLOG, we may assume that this interval is centered at $0$ since otherwise we can multiply by a suitable exponential to shift the Fourier support to be around $0$. We prove the following statement:

**Lemma D.1.** *Let $0 < \epsilon < 0.1$ be a parameter and let $l \geq \lceil \log 1/\epsilon \rceil$ be some parameter. Let $\mathcal{M}$ be a function such that $\widehat{\mathcal{M}}$ is supported on $[-l, l]$ and such that $\|\widehat{\mathcal{M}}\|_1 \leq 1$. Also assume that we have access to a function $f$ such that*

$$\int_{-1}^1 |f(x) - \mathcal{M}(x)|^2 dx \leq \epsilon^2 \,.$$

*There is an algorithm that runs in $\mathrm{poly}(l)$ time and outputs a function $\widetilde{\mathcal{M}}$ such that $\widetilde{\mathcal{M}}$ is $(O(l), O(l))$-simple, has Fourier support contained in $[-l, l]$, and*

$$\int_{-1}^1 |\widetilde{\mathcal{M}}(x) - f(x)|^2 dx \leq 16\epsilon^2 \,.$$

**Remark.** *Note that in this case, we do not need any constraint on the Fourier sparsity of $\mathcal{M}$ to guarantee that the output of our algorithm is $O(\log 1/\epsilon)$-Fourier sparse. Also, unlike our full result, Theorem 1.3, our output in this case is guaranteed to be a good approximation over the entire interval (instead of a subinterval).*

Our proof will be separated into two parts. The first step will be proving the existence of a function $\widetilde{\mathcal{M}}$ of the desired form. The second step will be developing an algorithm to actually compute it.

### D.0.1  Existence of a Sparse Approximation

First, we will prove that under the assumptions of Lemma D.1, an approximation $\widetilde{\mathcal{M}}$ satisfying the desired properties exists. We will also prove that independent of the problem instance, it suffices to only consider a fixed set of $O(l)$ distinct frequencies given by the Chebyshev points (with suitable rescaling).

The proof relies on first taking the Taylor series of an exponential $e^{2\pi i \zeta x}$ and arguing that we only need to keep the first $O(\log 1/\epsilon)$ terms. This essentially lets us represent such an exponential with the coefficients of its Taylor series, which are (up to rescaling) $(1, \zeta, \zeta^2, \dots)$. We then use Corollary F.7 to argue that an arbitary linear combination of such vectors can be replaced with a sparse combination with similarly sized coefficients.

**Lemma D.2.** *Let $0 < \epsilon < 0.1$ be a parameter and let $l \geq \lceil \log 1/\epsilon \rceil$ be some parameter. Let $t_0, \dots, t_{10^2 l}$ be the degree-$10^2 l$ Chebyshev points. Let $\mathcal{M}$ be a function such that $\widehat{\mathcal{M}}$ is supported on $[-l, l]$ and such that $\|\widehat{\mathcal{M}}\|_1 \leq 1$. Then there is a function*

$$h(x) = \sum_{j=0}^{10^2 l} c_j e^{2\pi i (l t_j) x}$$

*where $c_0, \dots, c_{10^2 l}$ are complex numbers such that $\sum_{j=0}^{10^2 l} |c_j| \leq 200l$ and*

$$\int_{-1}^1 (h(x) - \mathcal{M}(x))^2 dx \leq \epsilon^2 \,.$$

*Proof.* Note that $\mathcal{M}$ can be written as $\mathcal{M}(x) = \int_{-l}^l \widehat{\mathcal{M}}(\zeta) e^{2\pi i x \zeta} d\zeta$. Now consider the Taylor expansion of

$$e^{2\pi i x \zeta} = \sum_{j=0}^\infty \frac{(2\pi i x \zeta)^j}{j!}$$

Note that since $-l \leq \zeta \leq l$, we have

$$\sum_{j=10^2l+1}^{\infty} \left| \frac{(2\pi i \zeta)^j}{j!} \right| \leq (\epsilon/l)^3 \, .$$

In particular, if we define

$$g_\zeta(x) = \sum_{j=0}^{10^2l} \frac{(2\pi i x \zeta)^j}{j!}$$

then over the interval $x \in [-1, 1]$

$$|e^{2\pi i x \zeta} - g_\zeta(x)| \leq (\epsilon/l)^3 \, . \tag{10}$$

Next, for each $\zeta \in [-l, l]$, by Corollary F.7, we can write the vector

$$\mathcal{V}_{10^2l}(\zeta) = w_0(\zeta)\mathcal{V}_{10^2l}(t_0 l) + \cdots + w_{10^2l}(\zeta)\mathcal{V}_{10^2l}(t_{10^2l}l)$$

for some real numbers (depending on $\zeta$) $w_0(\zeta), \ldots, w_{10l}(\zeta)$ with $\sum |w_j(\zeta)| \leq 200l$. Thus,

$$g_\zeta(x) = w_0(\zeta)g_{t_0l}(x) + \cdots + w_{10^2l}(\zeta)g_{t_{10^2l}l}(x)$$

for the same weights. Now note that by (10), for all $x \in [-1, 1]$,

$$\left| \mathcal{M}(x) - \sum_{j=0}^{10^2l} g_{t_jl}(x) \left( \int_{-l}^{l} \widehat{\mathcal{M}}(\zeta)w_j(\zeta)d\zeta \right) \right| = \left| \mathcal{M}(x) - \int_{-l}^{l} \widehat{\mathcal{M}}(\zeta)g_\zeta(x)d\zeta \right| \leq 2l \cdot (\epsilon/l)^3 \, .$$

Note that

$$\sum_{j=0}^{10^2l} \left| \int_{-l}^{l} \widehat{\mathcal{M}}(\zeta)w_j(\zeta)d\zeta \right| \leq 200l$$

so by (10), for all $x \in [-1, 1]$,

$$\left| \sum_{j=0}^{10^2l} \left( g_{t_jl}(x) - e^{2\pi i x(t_jl)} \right) \left( \int_{-l}^{l} \widehat{\mathcal{M}}(\zeta)w_j(\zeta)d\zeta \right) \right| \leq 200l(\epsilon/l)^3$$

so therefore for all $x \in [-1, 1]$, we have

$$\left| \mathcal{M}(x) - \sum_{j=0}^{10^2l} e^{2\pi i x(t_jl)} \left( \int_{-l}^{l} \widehat{\mathcal{M}}(\zeta)w_j(\zeta)d\zeta \right) \right| \leq 202l \cdot (\epsilon/l)^3$$

and setting

$$h(x) = \sum_{j=0}^{10^2l} e^{2\pi i x(t_jl)} \left( \int_{-l}^{l} \widehat{\mathcal{M}}(\zeta)w_j(\zeta)d\zeta \right)$$

immediately leads to the desired conclusion. ∎

### D.0.2 Completing the Proof of Lemma D.1

By combining Lemma D.2 and Lemma F.10, we can complete the proof of Lemma D.1.

*Proof of Lemma D.1.* We can separate $f$ into its real and imaginary parts, say $f_{\mathsf{re}}, f_{\mathsf{im}}$ and we can separate $\mathcal{M}$ into its real and imaginary parts $\mathcal{M}_{\mathsf{re}}, \mathcal{M}_{\mathsf{im}}$. Now consider the Chebyshev points of degree $10^2l$, say $t_0, \ldots, t_{10^2l}$. We will now apply Lemma F.10 where we consider the set of functions

$$\{f_1, \ldots, f_n\} = \{\pm \cos(2\pi t_0 x), \pm \sin(2\pi t_0 x), \ldots, \pm \cos(2\pi t_{10^2l}x), \pm \sin(2\pi t_{10^2l}x)\} \, .$$

The distribution $\mathcal{D}$ is the uniform distribution on $[-1, 1]$ and by Lemma D.2, there are coefficients $a_1, \ldots, a_n \geq 0$ with $a_1 + \cdots + a_n \leq O(l)$ (note we can split the complex coefficients $c_j$ into their real and imaginary parts and then split into positive and negative parts) such that

$$\int_{-1}^{1} (f_{\mathsf{re}}(x) - (a_1 f_1(x) + \cdots + a_n f_n(x)))^2 dx \leq 2 \int_{-1}^{1} (\mathcal{M}_{\mathsf{re}}(x) - (a_1 f_1(x) + \cdots + a_n f_n(x)))^2 dx$$

$$+ 2 \int_{-1}^{1} (f_{\mathsf{re}}(x) - \mathcal{M}_{\mathsf{re}}(x))^2 dx \leq 4\epsilon^2$$

and similar for the imaginary part of $f$. Applying Lemma F.10 to both the real and imaginary part (after rescaling by $1/(O(l))$), adding the results, and rewriting the trigonometric functions using complex exponentials (note the set $\{t_0, \ldots, t_{10^2 l}\}$ is symmetric around $0$ so we can do this) completes the proof. ∎

We can slightly extend Lemma D.1 to work even if we do not know the desired accuracy $\epsilon$ but only a lower bound on it. It suffices to run the algorithm for Lemma D.1 and repeatedly decrease the target accuracy until our algorithm fails to find the optimal accuracy within a constant factor.

**Corollary D.3.** *Let $l, \epsilon$ be parameters given to us such that $l \geq \lceil \log 1/\epsilon \rceil$. Let $\mathcal{M}$ be a function such that $\widehat{\mathcal{M}}$ is supported on $[-l, l]$ and $\|\widehat{\mathcal{M}}\|_1 \leq 1$. Assume that we have access to a function $g$ defined on $[-1, 1]$. There is an algorithm that runs in $\mathrm{poly}(l)$ time and outputs a function $\widetilde{\mathcal{M}}$ such that $\widetilde{\mathcal{M}}$ is $(O(l), O(l))$-simple, has Fourier support contained in $[-l, l]$, and*

$$\int_{-1}^{1} |\widetilde{\mathcal{M}}(x) - f(x)|^2 dx \leq 20 \left( \epsilon^2 + \int_{-1}^{1} |f(x) - \mathcal{M}(x)|^2 dx \right).$$

*Proof.* For a target accuracy $\gamma > \epsilon$ we run the algorithm in Lemma D.1 to get a function $\widetilde{\mathcal{M}}_\gamma(x)$. We then check whether

$$\int_{-1}^{1} |\widetilde{\mathcal{M}}_\gamma(x) - f(x)|^2 dx \leq 16\gamma^2.$$

Note that the above can be explicitly computed. If the above check passes, we then take $\gamma \leftarrow 0.99\gamma$. Taking the smallest $\gamma$ for which the above succeeds, the guarantee from Lemma D.1 ensures that we have a function $\widetilde{\mathcal{M}}$ such that

$$\int_{-1}^{1} |\widetilde{\mathcal{M}}(x) - f(x)|^2 dx \leq 20 \left( \epsilon^2 + \int_{-1}^{1} |f(x) - \mathcal{M}(x)|^2 dx \right).$$

It is clear that we run the routine from Lemma D.1 at most $O(l)$ times so we are done. ∎

## E  Sparse Fourier Reconstruction: Full Version

In this section, we complete the proof of our main result on sparse Fourier reconstruction, Theorem 1.3. The high-level outline of the proof is similar to the proof of Theorem 1.1. The key lemma that goes into the proof is stated below. At a high level, the lemma states that if we know roughly where the Fourier support of the unknown Fourier-sparse signal $\mathcal{M}$ is located, then we can successfully reconstruct it.

**Lemma E.1.** *Assume we are given $N, k, \epsilon, c$ with $0 < \epsilon < 0.1$. Let $l = \lceil \log kN/(\epsilon c) \rceil$ be some parameter. Let $\mathcal{M}$ be a function that is $(k, 1)$-simple. Also, assume that we are given a set $T \subset \mathbb{R}$ of size $N$ such that all of the support of $\widehat{\mathcal{M}}$ is within distance $1$ of $N$. Further, assume we are given access to a function $f$ such that*

$$\int_{-1}^{1} |f(x) - \mathcal{M}(x)|^2 dx \leq \epsilon^2.$$

*There is an algorithm that runs in $\mathrm{poly}(N, k, l, 1/c)$ time and outputs a function $\widetilde{\mathcal{M}}$ that is $(k\mathrm{poly}(l/c), k\mathrm{poly}(l/c))$-simple and*

$$\int_{-1+c}^{1-c} |\widetilde{\mathcal{M}}(x) - f(x)|^2 dx \leq \epsilon^2 \mathrm{poly}(l).$$

The proof of Lemma E.1 will involve localizing the frequencies and then using Corollary D.3 to reconstruct after localizing. We will do this for $\mathrm{poly}(N, k, \log 1/\epsilon)$ different localizations (based on the set $T$ that we are given). We will then select at most $\widetilde{O}(k)$ of these localized reconstructions to add together and output. The intuition behind why we can find such a set of $\widetilde{O}(k)$ localized reconstructions and ignore the rest is that $\mathcal{M}$ is $k$-Fourier sparse so localizations that are far away from the frequencies of $\mathcal{M}$ can essentially be ignored.

The localization procedure will involve convolving $f$ by a Gaussian times an exponential (technically we will convolve by a function that approximates a Gaussian times an exponential). Note that this is equivalent to multiplying the Fourier transform by a Gaussian multiplier. This will ensure that frequencies too far away from a certain target frequency will only contribute negligibly and we only need to worry about reconstructing the frequencies that are close to the target frequency.

## E.1  Properties of Localization

In this section, we formalize the localization step and prove several inequalities that will be used in the proof of Lemma E.1. The way we would like to localize the frequencies is by multiplying by a Gaussian multiplier in Fourier space since afterwards, we would be able to essentially neglect any frequencies that are far away from the center of the Gaussian multiplier. This is equivalent to convolving by a Gaussian times an exponential, i.e. a function of the form $G(x)e^{2\pi i\theta x}$, in real space. For technical reasons, we will actually define two types of functions, which we call kernels, that approximate functions of the form $G(x)e^{2\pi i\theta x}$. The reason that we will need to work with both is that the first type can be computed efficiently while the second is easier to use in the analysis of our algorithm.

We begin with a few definitions.

**Definition E.2.** *For a function $f : \mathbb{R} \to \mathbb{C}$ and any $l > 0$, define $f^{trunc(l)}$ to be the function that is equal to $f$ on $[-l, l]$ and $0$ otherwise.*

**Definition E.3.** *For parameters $\mu, l, c$. we define the function*

$$\mathcal{K}_{\mu,l,c} = (1/c)\mathcal{P}_l^{trunc(l)}(x/c) \cdot e^{2\pi i\mu x}$$

*where $\mathcal{P}_l$ is as defined in Definition F.9.*

**Remark.** *We call functions of the above form truncated polynomial kernels. The fact that such functions are truncated polynomials will make it easy to explicitly compute convolutions.*

**Definition E.4.** *For parameters $\mu, l, a$, we define the function $\mathcal{T}_{\mu,l,a}$ as follows. First define*

$$\mathcal{S}_{l,a} = M_{0,1}^{trunc(l)}(x/a)$$

*(recall Definition B.1) and then define*

$$\mathcal{T}_{\mu,l,a}(x) = \widehat{\mathcal{S}_{l,a}}(x)e^{2\pi i\mu x}\,.$$

**Remark.** *We call functions of the above form truncated Gaussian kernels. Note that truncated Gaussian kernels are compactly supported in Fourier space, which will be a convenient property in the analysis of our algorithm later on.*

Note that both $\mathcal{K}_{0,l,c}$ and $\mathcal{T}_{0,l,2\pi/c}$ are meant to approximate the Gaussian $N(0, c^2)$. The fact that $\mathcal{K}_{0,l,c}$ approximates $N(0, c^2)$ is clear from the definition (and Lemma F.8). To see why $\mathcal{T}_{0,l,2\pi/c}$ approximates the Gaussian, note that if in the definition of $\mathcal{T}$, we did not truncate before taking the Fourier transform, then we would get exactly $N(0, c^2)$.

We will now prove several inequalities relating to how convolving with the kernels $\mathcal{K}$ and $\mathcal{T}$ affect a function. The first set of bounds are an immediate consequence of Lemma F.8.

**Claim E.5.** *Let $l, \epsilon > 0$ be parameters such that $l \geq \lceil \log 1/\epsilon \rceil$. Let $0 < c < 1$ be some constant. Then*

$$\|\mathcal{K}_{0,l,c}(x) - N(0, c^2)(x)\|_1 \leq O(\epsilon l)$$
$$\|\mathcal{K}_{0,l,c}(x) - N(0, c^2)(x)\|_2^2 \leq O(l\epsilon^2/c)\,.$$

*Proof.* We know by Lemma F.8 that

$$|(1/c)\mathcal{P}_l(x/c) - N(0, c^2)(x)| \le \epsilon/c$$

for all $x \in [-lc, lc]$. Thus, since $G$ decays rapidly outside the interval $[-lc, lc]$ we have

$$\|\mathcal{K}_{0,l,c}(x) - N(0, c^2)(x)\|_1 \le O(\epsilon l).$$

The second inequality follows by a similar argument. $\blacksquare$

The next claim formalizes the intuition that $\mathcal{K}_{\mu,l,c}$ and $\mathcal{T}_{\mu,l,2\pi/c}$ must be close because they both approximate the same function (the function $N(0, c^2)e^{2\pi i \mu x}$).

**Claim E.6.** *Let $l$ be some parameter and let $\epsilon > 0$ be such that $l \ge \lceil \log 1/\epsilon \rceil$. Let $g$ be a function such that $\|\widehat{g}\|_1 \le 1$. Then for any $\mu, c$,*

$$\|\mathcal{K}_{\mu,l,c} * g - \mathcal{T}_{\mu,l,2\pi/c} * g\|_\infty \le O(\epsilon l),$$

*where $*$ denotes convolution.*

*Proof.* First, note that it suffices to prove the above for $\mu = 0$. Now let $G = N(0, c^2)$. By Claim E.5, we have

$$\|\mathcal{K}_{0,l,c}(x) - G(x)\|_1 \le O(\epsilon l).$$

Since $\|\widehat{g}\|_1 \le 1$, we know that $\|g\|_\infty \le 1$ and thus for all $x$,

$$|\mathcal{K}_{0,l,c} * g(x) - G * g(x)| \le O(\epsilon l). \tag{11}$$

Finally, observe that the Fourier transform of $G * g$ is equal to $\widehat{G}\widehat{g}$. Note that

$$\widehat{G}(x) = M_{0,(2\pi/c)^2}(x) = M_{0,1}(cx/(2\pi)).$$

By construction,

$$\widehat{\mathcal{T}_{0,l,2\pi/c} * g} = M_{0,1}^{\mathsf{trunc}(l)}(cx/(2\pi))\widehat{g}$$

which is equal to $\widehat{G}\widehat{g}$ restricted to the interval $[-2\pi l/c, 2\pi l/c]$. Using the fact that $\widehat{G} = M_{0,(2\pi/c)^2}$ decays rapidly outside $[-2\pi l/c, 2\pi l/c]$, we have that

$$\|\widehat{\mathcal{T}_{0,l,2\pi/c} * g} - \widehat{G}\widehat{g}\|_1 \le \epsilon.$$

Thus, $|\mathcal{T}_{0,l,2\pi/c} * g(x) - G * g(x)| \le \epsilon$ for all $x$ and combining with (11), we are done. $\blacksquare$

## E.2 Decoupling

In our full algorithm, we will reconstruct frequency-localized versions of a function independently for different frequencies $\theta$ that we localize around. We will then combine our localized reconstructions by adding them. In this section, we prove several inequalities that will allow us to analyze what happens to our estimation error when we add different localized reconstructions together. Recall that convolving by a truncated polynomial kernel $\mathcal{K}_{\mu,l,c}$ or truncated Gaussian kernel $\mathcal{T}_{\mu,l,a}$ is approximately equivalent to multiplying the Fourier transform by a Gaussian multiplier centered around $\mu$. Lemma B.4 implies that adding up evenly spaced Gaussian multipliers approximates the constant function. Thus, we expect that convolving by an expression of the form $\sum_\mu \mathcal{K}_{\mu,l,c}$ or $\sum_\mu \mathcal{T}_{\mu,l,a}$ where the sum is over evenly spaced $\mu$ should roughly recover the original function. The first two claims here formalize this intuition.

In the first claim, we analyze what happens when we add several localizations obtained by convolving with various truncated polynomial kernels.

**Claim E.7.** *Let $l$ be some parameter and let $0 < \epsilon < 0.1$ be such that $l \ge \lceil \log 1/\epsilon \rceil$. Let $g$ be a function. Let $c$ be some constant. Let $S$ be a set of integer multiples of $2\pi/(cl)$. Then*

$$\int_{-1}^{1} \left| \frac{1}{l\sqrt{2\pi}} \sum_{\mu \in S} \mathcal{K}_{\mu,l,c} * g \right|^2 \le (1 + O(\epsilon|S|))^2 \int_{-(1+lc)}^{1+lc} |g|^2,$$

*Proof.* Note that $\mathcal{K}_{\mu,l,c}$ is supported on the interval $[-lc, lc]$ so we can restrict $g$ to be supported on $[-(1+lc), 1+lc]$ and 0 outside the interval. Define the function

$$V(x) = \frac{1}{l\sqrt{2\pi}} \sum_{\mu \in S} \mathcal{K}_{\mu,l,c}.$$

Then

$$\widehat{V}(x) = \frac{1}{l\sqrt{2\pi}} \sum_{\mu \in S} \widehat{\mathcal{K}_{0,l,c}}(x - \mu).$$

Next, let $G$ denote the Gaussian $G = N(0, c^2)$. Recall by Claim E.5, $\|\mathcal{K}_{0,l,c}(x) - G(x)\|_1 \leq O(\epsilon l)$ so $\|\widehat{\mathcal{K}_{0,l,c}} - \widehat{G}\|_\infty \leq O(\epsilon l)$. Let

$$U(x) = \frac{1}{l\sqrt{2\pi}} \sum_{\mu \in S} \widehat{G}(x - \mu).$$

Note that since $\widehat{G} = M_{0,(2\pi/c)^2}$, for any $x$,

$$|U(x)| = \frac{1}{l\sqrt{2\pi}} \sum_{\mu \in S} M_{0,(2\pi/c)^2}(x - \mu) \leq \frac{1}{l\sqrt{2\pi}} \sum_{\mu \in (2\pi)/(cl)\mathbb{Z}} M_{0,(2\pi/c)^2}(x - \mu) \leq 1 + \epsilon$$

where the last inequality follows from Lemma B.4. Also $\|\widehat{K_{0,l,c}} - \widehat{G}\|_\infty \leq O(\epsilon l)$ so for all $x$,

$$|\widehat{V}(x)| \leq 1 + O(\epsilon|S|).$$

We conclude

$$\int_{-1}^{1} |V * g(x)|^2 dx \leq \|V * g\|_2^2 = \|\widehat{V}\widehat{g}\|_2^2 \leq (1 + O(\epsilon|S|))^2 \|\widehat{g}\|_2^2 = (1 + O(\epsilon|S|))^2 \|g\|_2^2$$

To complete the proof recall that we restricted $g$ to be supported on $[-(1+lc), 1+lc]$ and we are done. ∎

The next claim is similar to the previous one except we analyze what happens when we add several localizations obtained by convolving with various truncated Gaussian kernels.

**Claim E.8.** *Let $a, l, \epsilon$ be parameters such that $0 < \epsilon < 0.1$ and $l \geq \lceil \log 1/\epsilon \rceil$. Let $g$ be a function whose Fourier support is contained in a set $S_0 \subset \mathbb{R}$ and such that $\|\widehat{g}\|_1 \leq 1$. Let $S$ be a set of integer multiples of $a/l$ that contains all multiples within a distance $l \cdot a$ of $S_0$. Then*

$$\|g - \frac{1}{l\sqrt{2\pi}} \sum_{\mu \in S} \mathcal{T}_{\mu,l,a} * g\|_\infty \leq 2\epsilon.$$

*Proof.* Consider the function

$$A(x) = \sum_{\mu \in S} \frac{1}{l\sqrt{2\pi}} M_{\mu,a^2}(x).$$

For all $x \in S_0$, we claim that

$$|A(x) - 1| \leq \epsilon.$$

To see this, note that by Lemma B.4,

$$\left| \sum_{\mu \in (a/l)\mathbb{Z}} \frac{1}{l\sqrt{2\pi}} M_{\mu,a^2}(x) - 1 \right| \leq 0.1\epsilon$$

for all $x$. By assumption, the set $S$ contains all integers that are within $la$ of the set $S_0$ so for any $x \in S_0$,

$$\sum_{\mu \in (a/l)\mathbb{Z}\setminus S} \frac{1}{l\sqrt{2\pi}} M_{\mu,a^2}(x) \leq 0.1\epsilon,$$

and we conclude that we must have $|A(x) - 1| \le \epsilon$. Next, we claim that if we define

$$B(x) = \sum_{\mu \in S} \frac{1}{l\sqrt{2\pi}} M_{0,1}^{\mathsf{trunc}(l)}((x - \mu)/a) \,,$$

then we have for all $x$,

$$|B(x) - A(x)| \le \epsilon \,.$$

To see this, first note that $M_{0,1}((x - \mu)/a) = M_{\mu,a^2}(x)$. Next, using Gaussian tail decay, we have for all $x$,

$$|B(x) - A(x)| \le \sum_{\substack{\mu \in (a/l)\mathbb{Z} \\ |x - \mu| \ge la}} \frac{1}{l\sqrt{2\pi}} M_{\mu,a^2}(x) \le \epsilon \,.$$

Thus, we have

$$|B(x) - 1| \le 2\epsilon$$

for $x \in S_0$. Note that by definition,

$$\widehat{g}(x) - \frac{1}{l\sqrt{2\pi}} \sum_{\mu \in S} \widehat{\mathcal{T}_{\mu,l,a} * g}(x) = \widehat{g}(x) - \widehat{g}(x) \sum_{\mu \in S} \frac{1}{l\sqrt{2\pi}} M_{0,1}^{\mathsf{trunc}(l)}((x - \mu)/a) = (1 - B(x))\widehat{g}(x)$$

so therefore

$$\left\| \widehat{g} - \frac{1}{l\sqrt{2\pi}} \sum_{\mu \in S} \widehat{\mathcal{T}_{\mu,l,a} * g} \right\|_1 \le 2\epsilon$$

and we conclude

$$\left\| g - \frac{1}{l\sqrt{2\pi}} \sum_{\mu \in S} \mathcal{T}_{\mu,l,a} * g \right\|_\infty \le 2\epsilon$$

as desired. $\blacksquare$

The last result in this section will allow us to decouple errors from summing over different localizations. Note that naively, if we add together $n$ estimates with $L^2$ errors $\epsilon_1, \ldots, \epsilon_n$, then the resulting $L^2$ error of the sum could be as large as $\epsilon_1 + \cdots + \epsilon_n$. If the estimates were "independent" on the other hand, we would expect the $L^2$ error of the sum to only be $\sqrt{\epsilon_1^2 + \cdots + \epsilon_n^2}$. We prove that when adding together functions that are frequency-localized at different locations, the error essentially matches the latter bound (up to logarithmic factors). This tighter bound will be necessary in the proof of Lemma E.1.

Note that if we have functions $g_1, \ldots, g_n$ whose Fourier supports are disjoint, then it is immediate that

$$\|g_1 + \cdots + g_n\|_2^2 = \|g_1\|_2^2 + \cdots + \|g_n\|_2^2 \,.$$

However, in our setting, we need to restrict the functions to the interval $[-1, 1]$ first, which causes the Fourier supports to no longer be disjoint. Through a few additional arguments we are able to prove an analogous statement for bounding $\int_{-1}^{1} |g_1 + \cdots + g_n|^2$. We do pay some additional losses both in the inequality itself and the bounds of the integral i.e. we need to integrate the individual functions over a slightly larger interval.

**Claim E.9.** *Let $\alpha, l, \epsilon$ be parameters such that $\alpha > 1$ and $l \ge \lceil \log \alpha n/\epsilon \rceil$. Let $I_1, \ldots, I_n$ be intervals of length at least $\alpha$ and assume that for any $x \in \mathbb{R}$, at most $l$ of the intervals contain $x$. Let $g_1, \ldots, g_n$ be functions such that for all $j \in [n]$, $\|\widehat{g_j}\|_1 \le 1$ and $\widehat{g_j}$ is supported on $I_j$. Then*

$$\int_{-1}^{1} |g_1 + \cdots + g_n|^2 \le \mathsf{poly}(l) \left( \epsilon^5 + \int_{-(1+\alpha^{-1})}^{(1+\alpha^{-1})} |g_1|^2 + \cdots + \int_{-(1+\alpha^{-1})}^{(1+\alpha^{-1})} |g_n|^2 \right) \,.$$

*Proof.* Consider the Gaussian multiplier $M = M_{\mu,\alpha^{-2}l^{-100}}$ for some $\mu \in [-1, 1]$. Now first, we bound

$$\int_{-\infty}^{\infty} M(x)^2 |g_1(x) + \cdots + g_n(x)|^2 dx \,.$$

Define the functions $h_j = \widehat{Mg_j}$ for all $j$. Then by Plancherel,

$$\int_{-\infty}^{\infty} M(x)^2 |g_1(x) + \cdots + g_n(x)|^2 dx = \int_{-\infty}^{\infty} |h_1 + \cdots + h_n|^2 dx \tag{12}$$

On the other hand, note that $h_j = \widehat{M} * \widehat{g_j}$. Let $J_j$ denote the interval containing all points within distance at most $10\alpha l^{60}$ of the interval $I_j$. Let $h'_j$ be the function $h_j$ restricted to $J_j$ (and equal to 0 outside). Recall that the support of $\widehat{g_j}$ is contained within $I_j$. Then we claim that

$$\int_{-\infty}^{\infty} |h_j - h'_j|^2 \le (\epsilon/(\alpha n))^{100} .$$

This follows because $|\widehat{M}| = N(0, (2\pi\alpha l^{50})^2)$ and $\|\widehat{g_j}\|_1 \le 1$ so for a point $x$ that is distance $d$ away from the interval $I_j$, we have

$$|h_j(x)| \le \max_{y \in I_j} |\widehat{M}(x - y)| \le N(0, (2\pi\alpha l^{50})^2)(d) .$$

Also note that $\|\widehat{g_j}\|_1 \le 1$, implies $\|g_j\|_\infty \le 1$ so

$$\|h_j\|_2^2 = \|Mg_j\|_2^2 \le \|M\|_2^2 \le 1 .$$

Combining the previous two inequalities over all $j$, we have

$$|\|h_1 + \cdots + h_n\|_2 - \|h'_1 + \cdots + h'_n\|_2| \le (\epsilon/(\alpha n))^{49}$$

$$\|h_1 + \cdots + h_n\|_2 + \|h'_1 + \cdots + h'_n\|_2 \le 3n$$

which implies

$$\int_{-\infty}^{\infty} |h_1 + \cdots + h_n|^2 dx \le 0.1(\epsilon/(\alpha n))^{10} + \int_{-\infty}^{\infty} |h'_1 + \cdots + h'_n|^2 dx . \tag{13}$$

Now note that since not too many of the intervals $I_j$ may contain the same point $x \in \mathbb{R}$, not too many of the extended intervals $J_j$ can contain the same point $x \in \mathbb{R}$. In particular, at most $O(l^{70})$ of the extended intervals can contain the same point $x \in \mathbb{R}$. In other words, each point $x \in \mathbb{R}$ is in the support of at most $O(l^{70})$ of the $h'_1, \ldots, h'_n$. Thus, by Cauchy Schwarz,

$$\int_{-\infty}^{\infty} |h'_1 + \cdots + h'_n|^2 \le O(l^{70}) \left( \int_{-\infty}^{\infty} |h'_1|^2 + \cdots + \int_{-\infty}^{\infty} |h'_n|^2 \right) . \tag{14}$$

Now we bound

$$\int_{-\infty}^{\infty} |h'_j|^2 \le \int_{-\infty}^{\infty} |h_j|^2 = \int_{-\infty}^{\infty} M(x)^2 |g_j(x)|^2 dx \le 0.1(\epsilon/(\alpha n))^{10} + \int_{-(1+\alpha^{-1})}^{1+\alpha^{-1}} M(x)^2 |g_j(x)|^2 dx \tag{15}$$

where the last step holds because $\|g_j\|_\infty \le 1$ and the multiplier $M(x)$ is always at most 1 and decays rapidly outside the interval $[-(1 + \alpha^{-1}), 1 + \alpha^{-1}]$ since $\mu \in [-1, 1]$. Putting everything together (12, 13, 14, 15), we get

$$\int_{-\infty}^{\infty} M(x)^2 |g_1(x) + \cdots + g_n(x)|^2 dx \le \text{poly}(l) \left( (\epsilon/(\alpha n))^5 + \sum_{j=1}^{n} \int_{-(1+\alpha^{-1})}^{1+\alpha^{-1}} M(x)^2 |g_j(x)|^2 dx \right) .$$

Now summing the above over different multipliers $M = M_{\mu, \alpha^{-2} l^{-100}}$ i.e. with $\mu$ uniformly spaced on $[-1, 1]$ with spacing $\alpha^{-1} l^{-50}$, we conclude

$$\int_{-1}^{1} |g_1(x) + \cdots + g_n(x)|^2 dx \le 10 \sum_{\mu} \int_{-1}^{1} M_{\mu, \alpha^{-2} l^{-100}}(x)^2 |g_1(x) + \cdots + g_n(x)|^2 dx$$

$$\le \text{poly}(l)\epsilon^5 + \text{poly}(l) \left( \sum_{\mu} \int_{-(1+\alpha^{-1})}^{(1+\alpha^{-1})} M_{\mu, \alpha^{-2} l^{-100}}(x)^2 (|g_1|^2 + \cdots + |g_n|^2) \right)$$

$$\le \text{poly}(l)\epsilon^5 + \text{poly}(l) \left( \int_{-(1+\alpha^{-1})}^{(1+\alpha^{-1})} |g_1|^2 + \cdots + \int_{-(1+\alpha^{-1})}^{(1+\alpha^{-1})} |g_n|^2 \right) .$$

∎

### E.2.1 Completing the Proof of Lemma E.1

In this section, we will complete the proof of Lemma E.1. First, we need to introduce some notation. We will carry over all of the notation from the statement of Lemma E.1. We also use the following conventions:

- Let $S_0 = \{\theta_1, \ldots, \theta_k\}$ be the frequencies in the Fourier support of $\mathcal{M}$
- Let $\gamma > 0$ be parameter to be chosen later and let $l' = \lceil \log 1/\gamma \rceil$ (we will ensure $\gamma$ is sufficiently small i.e. $\gamma < (\epsilon c/(kN))^K$ for some sufficiently large absolute constant $K$ )
- Let the function $r(x)$ be defined as $r(x) = f(x) - \mathcal{M}(x)$ on the interval $[-1, 1]$ and $r(x) = 0$ outside the interval.

Recall that the way we will reconstruct the function is by attempting to localize around each of the points in the given set $T$ and reconstructing the localized function using Corollary D.3. We then find $k\,\mathrm{poly}(l'/c)$ of these localized reconstructions that we can combine to approximate the entire function.

In the first claim, we bound the error of our reconstruction using Corollary D.3 for a given localization. Recall the two types of kernels, the truncated polynomial kernel and the truncated Gaussian kernel, defined in Section E.1. Consider the kernels $\mathcal{K}_{\mu,l',c/l'}$ and $\mathcal{T}_{\mu,l',2\pi l'/c}$ (which, recall, are approximately the same). In the next lemma, we will bound the distance between $\mathcal{K}_{\mu,l',c/l'} * f$ and $\mathcal{T}_{\mu,l',2\pi l'/c} * \mathcal{M}$ in terms of $r(x)$. The reason we care about these two functions is that the first is something that we can compute since we are given explicit access to $f$. On the other hand, the second is Fourier sparse and has bounded Fourier support so it can be plugged into Corollary D.3 (as the unknown function $\mathcal{M}$).

**Claim E.10.** *For any real number $\mu$,*

$$\int_{-1+c}^{1-c} |\mathcal{K}_{\mu,l',c/l'} * f - \mathcal{T}_{\mu,l',2\pi l'/c} * \mathcal{M}|^2 \le \mathrm{poly}(l'/c)\gamma^2 + 4\int_{-\infty}^{\infty} |M_{0,(2\pi l'/c)^2}(x-\mu)\widehat{r}(x)|^2 dx\,.$$

*Proof.* Note that since $\mathcal{K}_{\mu,l',c/l'}$ is supported on $[-c, c]$,

$$\int_{-1+c}^{1-c} \left| \left(\mathcal{K}_{\mu,l',c/l'} * f\right)(x) - \left(\mathcal{K}_{\mu,l',c/l'} * \mathcal{M}\right)(x) \right|^2 dx \le \int_{-1}^{1} |\left(\mathcal{K}_{\mu,l',c/l'} * r\right)(x)|^2 dx$$

Now the Fourier transform of $\mathcal{K}_{\mu,l',c/l'} * r$ is $\widehat{\mathcal{K}_{0,l',c/l'}}(x-\mu)\widehat{r}(x)$ so

$$\int_{-1}^{1} |\left(\mathcal{K}_{\mu,l',c/l'} * r\right)(x)|^2 dx \le \int_{-\infty}^{\infty} |\widehat{\mathcal{K}_{0,l',c/l'}}(x-\mu)\widehat{r}(x)|^2 dx$$

We deduce that

$$\int_{-1+c}^{1-c} |\mathcal{K}_{\mu,l',c/l'} * f - \mathcal{T}_{\mu,l',2\pi l'/c} * \mathcal{M}|^2 \le 2 \int_{-1+c}^{1-c} \left| \mathcal{K}_{\mu,l',c/l'} * \mathcal{M} - \mathcal{T}_{\mu,l',2\pi l'/c} * \mathcal{M}\right|^2$$

$$+ 2 \int_{-\infty}^{\infty} |\widehat{\mathcal{K}_{0,l',c/l'}}(x-\mu)\widehat{r}(x)|^2 dx$$

$$\le \mathrm{poly}(l'/c)\gamma^2 + 2 \int_{-\infty}^{\infty} |\widehat{\mathcal{K}_{0,l',c/l'}}(x-\mu)\widehat{r}(x)|^2 dx$$

where the last inequality follows from Claim E.6.

Note since $r$ is supported on $[-1, 1]$ and $\|r\|_2^2 \le \epsilon^2 \le 0.1$, we must have $\|r\|_1 \le 1$ which then implies $\|\widehat{r}\|_\infty \le 1$. Together with Claim E.5, if we let $G = N(0, (c/l')^2)$ then we have

$$\int_{-\infty}^{\infty} |\widehat{\mathcal{K}_{0,l',c/l'}}(x-\mu)\widehat{r}(x)|^2 dx \le 2 \int_{-\infty}^{\infty} |\widehat{G}(x-\mu)\widehat{r}(x)|^2 dx + 2\|\widehat{r}\|_\infty^2 \|\mathcal{K}_{0,l',c/l'} - G\|_2^2$$

$$\le \mathrm{poly}(l'/c)\gamma^2 + 2 \int_{-\infty}^{\infty} |M_{0,(2\pi l'/c)^2}(x-\mu)\widehat{r}(x)|^2 dx\,.$$

and combining with the previous inequality, we get the desired result. ∎

We are now ready to complete the proof of Lemma E.1.

*Proof of Lemma E.1.* First, let $T'$ be the set of integer multiples of $2\pi/c$ that are within distance $(10l')^2/c$ of the set $T$. For all $\mu \in T'$, do the following. We compute the function $f^{(\mu)} = \mathcal{K}_{\mu,l',c/l'} * f$. Next we apply Corollary D.3 (with appropriate rescaling) to compute a function $h^{(\mu)}$ in $\mathrm{poly}(l'/c)$ time that has Fourier support in $[\mu - 2\pi l'^2/c, \mu + 2\pi l'^2/c]$, is $(\mathrm{poly}(l'/c), \mathrm{poly}(l'/c))$-simple and such that

$$\int_{-1+c}^{1-c} |f^{(\mu)} - h^{(\mu)}|^2 \leq 20 \left( \gamma^2 + \int_{-1+c}^{1-c} |f^{(\mu)} - \mathcal{T}_{\mu,l',2\pi l'/c} * \mathcal{M}|^2 \right). \tag{16}$$

To see why we can do this, note that

$$\widehat{\mathcal{T}_{\mu,l',2\pi l'/c} * \mathcal{M}} = M_{0,1}^{\mathrm{trunc}(l')}(c(x-\mu)/(2\pi l'))\widehat{\mathcal{M}}$$

is supported on $[\mu - 2\pi l'^2/c, \mu + 2\pi l'^2/c]$. Also it is clear that $\|\widehat{\mathcal{T}_{\mu,l',2\pi l'/c} * \mathcal{M}}\|_1 \leq \|\widehat{\mathcal{M}}\|_1 \leq 1$.

Now we choose a set $U \subset T'$ with $|U| \leq k(10l')^2$ such that the following quantity is minimized:

$$E_U = \sum_{\mu \in U} \int_{-1+c}^{1-c} |f^{(\mu)} - h^{(\mu)}|^2 + \sum_{\mu \in T' \backslash U} \int_{-1+c}^{1-c} |f^{(\mu)}|^2.$$

Note that this can be done using a simple greedy procedure. First, we obtain a bound on the value $E_U$ that we compute. Let $U_0$ be the set of all integer multiples of $2\pi/c$ that are within distance $10l'^2/c$ of $S_0$ (recall that $S_0$ is the Fourier support of $\mathcal{M}$ which consist of $k$ points). By assumption, we know that $U_0 \subset T'$ and it is clear that $|U_0| \leq k(10l')^2$. Note that by definition, for any $\mu \notin U_0$, the function $\mathcal{T}_{\mu,l',2\pi l'/c} * \mathcal{M}$ is identically 0. Now using (16), then Claim E.10,

$$E_{U_0} \leq 20 \sum_{\mu \in U_0} \left( \gamma^2 + \int_{-1+c}^{1-c} |f^{(\mu)} - \mathcal{T}_{\mu,l',2\pi l'/c} * \mathcal{M}|^2 \right) + \sum_{\mu \in T' \backslash U_0} \int_{-1+c}^{1-c} |f^{(\mu)}|^2$$

$$\leq 80 \sum_{\mu \in T'} \left( \mathrm{poly}(l'/c)\gamma^2 + \int_{-\infty}^{\infty} |M_{0,(2\pi l'/c)^2}(x-\mu)\widehat{r}(x)|^2 dx \right)$$

$$\leq |T'|\mathrm{poly}(l'/c)\gamma^2 + 80 \int_{-\infty}^{\infty} |\widehat{r}(x)|^2 \sum_{\mu \in (2\pi/c)\mathbb{Z}} M_{0,(2\pi l'/c)^2}(x-\mu)^2 dx$$

$$\leq \gamma + \mathrm{poly}(l')\|r\|_2^2.$$

Note that we used the fact that $\gamma$ is sufficiently small and the tail decay properties of the Gaussian multipliers in the last step. Thus, we can ensure that the error that we compute satisfies $E_U \leq \gamma + \mathrm{poly}(l')\|r\|_2^2$. Now we output the function

$$\widetilde{\mathcal{M}} = \sum_{\mu \in U} \frac{1}{l'\sqrt{2\pi}} h^{(\mu)}.$$

It remains to bound the error between $\widetilde{\mathcal{M}}$ and $f$. First we apply Claim E.9 to decouple over all $\mu \in T'$. Note that $h^{(\mu)}$ and $\mathcal{T}_{\mu,l',2\pi l'/c}*\mathcal{M}$ both have Fourier support contained in the interval $[\mu - 2\pi l'^2/c, \mu + 2\pi l'^2/c]$. For distinct $\mu$ that are integer multiples of $2\pi/c$, there are at most $O(l'^2)$ intervals that contain any point. Also, note that for all $\mu$, $\|h^{(\mu)}\|_1 \leq \mathrm{poly}(l'/c)$ and $\|\widehat{\mathcal{T}_{\mu,l',2\pi l'/c} * \mathcal{M}}\|_1 \leq 1$. Thus,

by Claim E.9 (with appropriate rescaling of the functions and the interval),

$$
\int_{-1+2c}^{1-2c}\left|\widetilde{\mathcal{M}} - \sum_{\mu\in T'}\frac{1}{l'\sqrt{2\pi}}\mathcal{T}_{\mu,l',2\pi l'/c} * \mathcal{M}\right|
$$

$$
\leq \operatorname{poly}(l'/c)\gamma + \operatorname{poly}(l')\left(\sum_{\mu\in U}\int_{-1+c}^{1-c}|h^{(\mu)} - \mathcal{T}_{\mu,l',2\pi l'/c} * \mathcal{M}|^2 + \sum_{\mu\in T'\backslash U}\int_{-1+c}^{1-c}|\mathcal{T}_{\mu,l',2\pi l'/c} * \mathcal{M}|^2\right)
$$

$$
\leq \operatorname{poly}(l'/c)\gamma + \operatorname{poly}(l')\left(\sum_{\mu\in U}\int_{-1+c}^{1-c}|h^{(\mu)} - f^{(\mu)}|^2 + \sum_{\mu\in T'\backslash U}\int_{-1+c}^{1-c}|f^{(\mu)}|^2\right)
$$

$$
+ \operatorname{poly}(l')\sum_{\mu\in T'}\int_{-1+c}^{1-c}|f^{(\mu)} - \mathcal{T}_{\mu,l',2\pi l'/c} * \mathcal{M}|^2
$$

$$
\leq \operatorname{poly}(l'/c)\gamma + \operatorname{poly}(l')\left(E_U + \sum_{\mu\in T'}\left(\operatorname{poly}(l'/c)\gamma^2 + \int_{-\infty}^{\infty}|M_{0,(2\pi l'/c)^2}(x-\mu)\widehat{r}(x)|^2 dx\right)\right)
$$

$$
\leq \operatorname{poly}(l'/c)\gamma + \operatorname{poly}(l')(E_U + \|r\|_2^2)
$$

$$
\leq \operatorname{poly}(l')\epsilon^2
$$

where we used that $\gamma = (\epsilon c/(kN))^{O(1)}$ is sufficiently small and Claim E.10 and the last two inequalities follow from the same argument as in the bound for $E_{U_0}$. Next, by Claim E.8 (and the definition of $T'$), we have that

$$
\left\|\mathcal{M} - \sum_{\mu\in T'}\frac{1}{l'\sqrt{2\pi}}\mathcal{T}_{\mu,l',2\pi l'/c} * \mathcal{M}\right\|_\infty \leq O(\epsilon)
$$

so overall, we conclude

$$
\int_{-1+2c}^{1-2c}|\widetilde{\mathcal{M}} - \mathcal{M}|^2 \leq \operatorname{poly}(l')\epsilon^2
$$

from which we immediately deduce

$$
\int_{-1+2c}^{1-2c}|\widetilde{\mathcal{M}} - f|^2 \leq \operatorname{poly}(l')\epsilon^2 \,.
$$

It is also clear that $\widetilde{\mathcal{M}}$ is $(k\operatorname{poly}(l'/c), k\operatorname{poly}(l'/c))$-simple (since it is a sum of at most $k(10l')^2$ functions that are $(\operatorname{poly}(l'/c), \operatorname{poly}(l'/c))$-simple). Now we are done because $l' = O(l)$. ∎

Similar to obtaining Corollary D.3 from Lemma D.1, we can extend Lemma E.1 to work even when we do not know the target accuracy but only a lower bound on it.

**Corollary E.11.** *Assume we are given $N, k, \epsilon, c$ with $0 < \epsilon < 0.1$. Let $l = \lceil \log kN/(\epsilon c) \rceil$ be some parameter. Let $\mathcal{M}$ be a function that is $(k, 1)$-simple. Also, assume that we are given a set $T \subset \mathbb{R}$ of size $N$ such that all of the support of $\widehat{\mathcal{M}}$ is within distance $1$ of $N$. Further, assume we are given access to a function $f$. There is an algorithm that runs in $\operatorname{poly}(N, k, l, 1/c)$ time and outputs a function $\widetilde{\mathcal{M}}$ that is $(k\operatorname{poly}(l/c), k\operatorname{poly}(l/c))$-simple and such that*

$$
\int_{-1+c}^{1-c}|\widetilde{\mathcal{M}}(x) - f(x)|^2 dx \leq \epsilon^2 + \operatorname{poly}(l)\int_{-1}^{1}|f(x) - \mathcal{M}(x)|^2 dx\,.
$$

*Proof.* This will be the exact same argument as the proof of Corollary D.3. For a target accuracy $\epsilon' > \epsilon$ we run the algorithm in Lemma E.1 to get a function $\widetilde{\mathcal{M}}_{\epsilon'}(x)$. We then check whether

$$
\int_{-1+c}^{1-c}|\widetilde{\mathcal{M}}_{\epsilon'}(x) - f(x)|^2 dx \leq \operatorname{poly}(l)\epsilon'^2 \,.
$$

If the check passes, we take $\epsilon' \leftarrow 0.99\epsilon'$ and repeat the above until we find the smallest $\epsilon'$ (up to a constant factor) for which the check passes. The guarantee of Lemma E.1 implies that for this $\epsilon'$, we can just output $\widetilde{\mathcal{M}}_{\epsilon'}(x)$ and it is guaranteed to satisfy the desired inequality. ∎

### E.3 Proof of Main Theorem

Using Theorem 2.12 and Corollary E.11, we can prove our main theorem, Theorem 1.3. The main thing that we need to prove is that the frequencies in the function $f'$ computed by Theorem 2.12 cover (within distance $\text{poly}(k, \log 1/\epsilon)$) all of the frequencies in $\mathcal{M}$. This will then let us use the frequencies in $f'$ to construct a set $T$ of size $\text{poly}(k, \log 1/\epsilon)$ that covers all frequencies in $\mathcal{M}$ to within distance 1 that we can then plug into Corollary E.11. We will need the following technical lemma from Chen et al. [2016].

**Lemma E.12.** *[Lemma 5.1 in Chen et al. [2016]] For any $k$-Fourier sparse signal $g : \mathbb{R} \to \mathbb{C}$,*

$$\max_{x \in [-1,1]} |g(x)|^2 \leq O(k^4 \log^3 k) \int_{-1}^{1} |g(x)|^2 dx \,.$$

The above lemma roughly says that the mass of a $k$-Fourier sparse function cannot be too concentrated. We now finish the proof of Theorem 1.3.

*Proof of Theorem 1.3.* We first apply Theorem 2.12 to compute a function $f'$. such that

1. $f'$ is $(\text{poly}(k, \log 1/\epsilon), \exp(\text{poly}(k, \log 1/\epsilon)))$- simple

2.
$$\int_{-1}^{1} |f' - f|^2 \leq O\left(\epsilon^2 + \int_{-1}^{1} |f - \mathcal{M}|^2\right) \,.$$

Let $L = (k \log 1/\epsilon)^K$ for some sufficiently large absolute constant $K$. Let $\gamma = e^{-L}$. Now apply Claim E.8 on the function $f'$ with parameters $a \leftarrow L, l \leftarrow L, \epsilon \leftarrow \gamma$. Let $S \subset L\mathbb{Z}$ be the set of all integer multiples of $L$ that are within distance $L^3$ of the Fourier support of $f'$. We have

$$\left\| f' - \frac{1}{L\sqrt{2\pi}} \sum_{\mu \in S} \mathcal{T}_{\mu, L, L^2} * f' \right\|_\infty \leq 2\|\widehat{f'}\|_1 \gamma \tag{17}$$

Next, we apply Claim E.6 on the function $(\mathcal{M} - f')$. We deduce that for any $\mu$,

$$\left\| \mathcal{K}_{\mu, L, 2\pi/L^2} * (\mathcal{M} - f') - \mathcal{T}_{\mu, L, L^2} * (\mathcal{M} - f') \right\|_\infty \leq O\left(\gamma L \left(\|\widehat{f'}\|_1 + \|\widehat{\mathcal{M}}\|_1\right)\right) \,. \tag{18}$$

Finally, by Claim E.7 applied to the function $(\mathcal{M} - f')$ (with parameters $\epsilon \leftarrow \gamma, l \leftarrow L, c \leftarrow (2\pi)/L^2$), we have

$$\int_{-1}^{1} \left| \frac{1}{L\sqrt{2\pi}} \sum_{\mu \in S} \mathcal{K}_{\mu, L, 2\pi/L^2} * (\mathcal{M} - f') \right|^2 \leq (1 + O(\gamma|S|))^2 \int_{-1-2\pi/L}^{1+2\pi/L} |\mathcal{M} - f'|^2 \leq 2 \int_{-1}^{1} |\mathcal{M} - f'|^2 \,.$$

Note that in the last step we use Lemma E.12 and the fact that $\mathcal{M} - f'$ is $\text{poly}(k, \log 1/\epsilon)$-Fourier sparse so choosing $L = (k \log 1/\epsilon)^{O(1)}$ sufficiently large ensures that

$$\int_{-1-2\pi/L}^{1+2\pi/L} |\mathcal{M} - f'|^2 \leq 1.1 \int_{-1}^{1} |\mathcal{M} - f'|^2 \,.$$

Define the functions

$$A(x) = f' - \frac{1}{L\sqrt{2\pi}} \sum_{\mu \in S} \mathcal{T}_{\mu, L, L^2} * \mathcal{M}$$

$$B(x) = \frac{1}{L\sqrt{2\pi}} \sum_{\mu \in S} \mathcal{K}_{\mu, L, 2\pi/L^2} * (\mathcal{M} - f')$$

Note $\|A\|_\infty, \|B\|_\infty \leq |S|\left(\|\widehat{f'}\|_1 + \|\widehat{\mathcal{M}}\|_1\right)$. Combining (17, 18) we have

$$\int_{-1}^{1} |A(x)|^2 - \int_{-1}^{1} |B(x)|^2 \leq \gamma \text{poly}\left(L, |S|, \|\widehat{f'}\|_1 + \|\widehat{\mathcal{M}}\|_1\right)$$

However, we proved that $\int_{-1}^{1} |B(x)|^2 \leq 2 \int_{-1}^{1} |\mathcal{M} - f'|^2$ so choosing $L = (k \log 1/\epsilon)^{O(1)}$ sufficiently large and since $\gamma = e^{-L}$, we conclude

$$\int_{-1}^{1} |A(x)|^2 \leq \epsilon^2 + 2 \int_{-1}^{1} |\mathcal{M} - f'|^2 \leq O\left(\epsilon^2 + \int_{-1}^{1} |\mathcal{M} - f|^2\right) \tag{19}$$

where we are using the guarantee from Theorem 2.12. Now note that the function

$$\mathcal{M}' = \frac{1}{L\sqrt{2\pi}} \sum_{\mu \in S} \mathcal{T}_{\mu, L, L^2} * \mathcal{M}$$

is $k$-Fourier sparse and has $\|\widehat{\mathcal{M}'}\|_1 \leq 2\|\widehat{\mathcal{M}}\|_1$ by Lemma B.4. Furthermore, all of its Fourier support is within distance $\mathrm{poly}(L)$ of the Fourier support of $f'$ (by the construction of the set $S$). Thus, we can apply Corollary E.11 on $f'$ (where we treat the unknown function as $\mathcal{M}'$) with $N = \mathrm{poly}(L) = \mathrm{poly}(k, \log 1/\epsilon)$ and recover a function $\widetilde{\mathcal{M}}$ such that

$$\int_{-1+c}^{1-c} |\widetilde{\mathcal{M}} - f'|^2 \leq \epsilon^2 + \mathrm{poly}(\log k/(\epsilon c)) \int_{-1}^{1} |f' - \mathcal{M}'| = \epsilon^2 + \mathrm{poly}(\log k/(\epsilon c)) \int_{-1}^{1} |A(x)|^2$$

$$= \mathrm{poly}(\log k/(\epsilon c)) \left(\epsilon^2 + \int_{-1}^{1} |\mathcal{M} - f|^2\right)$$

where the last step is from (19). Since $\int_{-1}^{1} |f' - f|^2 \leq O\left(\epsilon^2 + \int_{-1}^{1} |f - \mathcal{M}|^2\right)$, the above implies

$$\int_{-1+c}^{1-c} |\widetilde{\mathcal{M}} - f|^2 \leq \mathrm{poly}(\log k/(\epsilon c)) \left(\epsilon^2 + \int_{-1}^{1} |\mathcal{M} - f|^2\right)$$

and we are done. $\blacksquare$

### E.4    Implementation of Computations

In the proof of Theorem 1.3, we use the result from Chen et al. [2016] to obtain an approximation $f'$ that is written as a sum of $\mathrm{poly}(k, \log 1/\epsilon)$ exponentials and has coefficients bounded by $\exp(\mathrm{poly}(k, \log 1/\epsilon))$. We then perform explicit computations using this function in our algorithm to eventually compute a sparser approximation with smaller coefficients. Here we briefly explain why these explicit computations can all be implemented efficiently. Note that all of the functions that we perform computations on can be written as sums of polynomials multiplied by exponentials i.e.

$$P_1(x)e^{2\pi i \theta_1 x} + \cdots + P_n(x)e^{2\pi i \theta_n x} \tag{20}$$

where there are at most $\mathrm{poly}(k, \log 1/\epsilon)$ terms in the sum, all of the polynomials have degree at most $\mathrm{poly}(k, \log 1/\epsilon)$ and all of the coefficients are bounded by $\exp(\mathrm{poly}(k, \log 1/\epsilon))$. To see this, note that convolving by a polynomial $P(x)$ truncated to an interval (recall the truncated polynomial kernel in Definition E.3 ) preserves a function of the form in (20) (only increasing the degrees of the polynomials by $\deg(P)$). All other computations that we need such as computing the exact value, adding and multiplying and integrating over some interval can clearly be done explicitly in $\mathrm{poly}(k, \log 1/\epsilon)$ time and to $\exp(\mathrm{poly}(k, \log 1/\epsilon))^{-1}$ accuracy for functions of the form specified in (20).

## F    Basic Tools

In this section, we have a few basic tools that are used repeatedly throughout the paper.

### F.1    Chebyshev Polynomials

Here we will introduce several basic results about the Chebyshev polynomials, which have algorithmic applications in a wide variety of settings Rivlin [2020], Guruswami and Zuckerman [2016].

**Definition F.1** (Chebyshev Polynomials)**.** *The Chebyshev Polynomials are a family of polynomials defined as follows: $T_0(x) = 1, T_1(x) = x$ and for $n \geq 2$,*

$$T_n(x) = 2xT_{n-1}(x) - T_{n-2}(x).$$

**Fact F.2.** *The Chebyshev polynomials satisfy the following property:*

$$T_n(\cos\theta) = \cos n\theta\,.$$

As an immediate consequence of the above, we have a few additional properties.

**Fact F.3.** *The Chebyshev polynomials satisfy the following properties:*

1. $T_n(x)$ *has degree $n$ and leading coefficient $2^{n-1}$*

2. *For $x \in [-1, 1]$, $T_n(x) \in [-1, 1]$*

3. $T_n(x)$ *has $n$ zeros all in the interval $[-1, 1]$*

4. *There are $n + 1$ values of $x$ for which $T_n(x) = \pm 1$, all in the interval $[-1, 1]$*

In light of the previous properties, we make the following definition.

**Definition F.4.** *For an integer $n$, we define the Chebyshev points of degree $n$, say $t_0, \ldots, t_n$, as the points in the interval $[-1, 1]$ where the Chebyshev polynomial satisfies $T_n(t_j) = \pm 1$. Note that the Chebyshev points are exactly*

$$\left\{\cos 0, \cos \frac{\pi}{n}, \ldots \cos \frac{(n-1)\pi}{n}, \cos \pi\right\}.$$

Next, we have a result saying that if we have a bound on the value of a degree-$n$ polynomial at all of the degree $n$ Chebyshev points, then we can bound the value over the entire interval $[-1, 1]$. Similar results are used in Rivlin [2020], Guruswami and Zuckerman [2016], but there does not appear to be a directly usable reference.

**Claim F.5.** *Let $P(x)$ be a polynomial of degree at most $n$ with real coefficients. Let $t_0, \ldots, t_n$ be the Chebyshev points of degree $n$. Assume that $|P(t_j)| \leq 1$ for $j = 0, 1, \ldots, n$. Then $|P(x)| \leq 2n$ for all $x \in [-1, 1]$.*

*Proof.* By Lagrange interpolation, we may write

$$P(x) = \frac{P(t_0)(x - t_1)\cdots(x - t_n)}{(t_0 - t_1)\cdots(t_0 - t_n)} + \cdots + \frac{P(t_n)(x - t_0)\cdots(x - t_{n-1})}{(t_n - t_0)\cdots(t_n - t_{n-1})}\,.$$

Thus, it suffices to upper bound the quantity

$$F(x) = \left|\frac{(x - t_1)\cdots(x - t_n)}{(t_0 - t_1)\cdots(t_0 - t_n)}\right| + \cdots + \left|\frac{(x - t_0)\cdots(x - t_{n-1})}{(t_n - t_0)\cdots(t_n - t_{n-1})}\right|$$

on the interval $[-1, 1]$. Note that by Lagrange interpolation on $T_n(x)$, we have

$$T_n(x) = \frac{T_n(t_0)(x - t_1)\cdots(x - t_n)}{(t_0 - t_1)\cdots(t_0 - t_n)} + \cdots + \frac{T_n(t_n)(x - t_0)\cdots(x - t_{n-1})}{(t_n - t_0)\cdots(t_n - t_{n-1})}\,.$$

Also note that $T_n(t_j) = (-1)^{n-j}$ which has the same sign as $(t_j - t_0)\cdots(t_j - t_{j-1})(t_j - t_{j+1})\cdots(t_j - t_n)$. Thus,

$$\left|\frac{1}{(t_0 - t_1)\cdots(t_0 - t_n)}\right| + \cdots + \left|\frac{1}{(t_n - t_0)\cdots(t_n - t_{n-1})}\right| = 2^{n-1}\,,$$

since the leading coefficient of $T_n(x)$ is $2^{n-1}$. Now we will upper bound

$$M = \max\left(|(x - t_1)\cdots(x - t_n)|, \ldots, |(x - t_0)\cdots(x - t_{n-1})|\right)$$

and once we do this, we will have a bound on $F(x)$ since $F(x) \leq 2^{n-1}M$. Define the polynomial

$$Q(x) = (x - t_0)(x - t_1)\cdots(x - t_n) = \frac{\sqrt{x^2 - 1}}{2^n}\left((x + \sqrt{x^2 - 1})^n - (x - \sqrt{x^2 - 1})^n\right)\,.$$

To see why the last equality is true, note that the RHS has roots at $t_0, \ldots, t_n$ and is a monic polynomial of degree $n + 1$ so it must be equal to $(x - t_0)\cdots(x - t_n)$. Now,

$$M = \max\left(\left|\frac{Q(x)}{x - t_0}\right|, \ldots, \left|\frac{Q(x)}{x - t_n}\right|\right) \leq \max |Q'(x)|$$

where the last step holds by the mean value theorem (because $Q(t_j) = 0$ for all $j$). Now note that

$$Q(\cos\theta) = -\frac{\sin\theta\sin(n\theta)}{2^{n-1}}$$

so

$$Q'(\cos\theta) = \frac{n\cos n\theta}{2^{n-1}} + \frac{\cos\theta\sin(n\theta)}{\sin(\theta)2^{n-1}}$$

and from the above it is clear that

$$|Q'(\cos\theta)| \leq \frac{n}{2^{n-1}} + \frac{n}{2^{n-1}} = \frac{n}{2^{n-2}}.$$

Now we are done because

$$\max_{x\in[-1,1]}|P(x)| \leq F(x) \leq 2^{n-1}M \leq 2n.$$

∎

It turns out that we can restate the above result in terms of convex hulls of points on the moment curve. This reformulation is the version that is useful in our algorithms.

**Definition F.6.** *For a real number $x$, we define the moment vector $\mathcal{V}_n(x) = (1, x, \ldots, x^n)$.*

**Corollary F.7.** *Let $t_0, t_1, \ldots, t_n$ be the Chebyshev points of degree $n$. Then for any $x \in [-1, 1]$, the point $\mathcal{V}_n(x)$ is contained in the convex hull of the points*

$$\{\pm 2n\mathcal{V}_n(t_0), \ldots, \pm 2n\mathcal{V}_n(t_n)\}.$$

*Proof.* Assume for the sake of contradiction that the above is not true. Then there must be a separating hyperplane. Assume that this hyperplane is given by $a \cdot x = b$ where $a$ is a vector and $b$ is a real number. Now WLOG $b \geq 0$ and we must have

$$a \cdot \mathcal{V}_n(x) \geq b$$

$$|a \cdot \mathcal{V}_n(t_j)| \leq \frac{b}{2n} \quad \forall j$$

However, applying Claim F.5 with $P(x) = \frac{2n(a\cdot\mathcal{V}_n(x)))}{b}$ gives a contradiction. Thus, no separating hyperplane can exist and we are done. ∎

### F.2 Approximating a Gaussian with a Polynomial

We will also need to approximate Gaussians with polynomials. This is a somewhat standard result which we state below.

**Lemma F.8.** *Let $G = N(0, 1)$ be the standard Gaussian. Let $l$ be some parameter. Then we can compute a polynomial $P(x)$ of degree $(10l)^2$ such that for all $x \in [-2l, 2l]$,*

$$|G(x) - P(x)| \leq e^{-l}.$$

*Proof.* Write

$$G(x) = \frac{1}{\sqrt{2\pi}}e^{-x^2/2}.$$

and now we can write the Taylor expansion

$$e^{-\frac{x^2}{2}} = \sum_{m=0}^{\infty} \frac{\left(-\frac{x^2}{2}\right)^m}{m!} = \sum_{m=0}^{\infty} \frac{(-1)^m x^{2m}}{2^m m!}$$

Now define

$$P(x) = \sum_{m=0}^{(10l)^2} \frac{(-1)^m x^{2m}}{2^m m!}.$$

For $x \in [-2l, 2l]$, we have

$$|G(x) - P(x)| \le \left| \sum_{m=(10l)^2+1}^{\infty} \frac{(-1)^m x^{2m}}{2^m m!} \right| \le \sum_{m=10^2 l+1}^{\infty} \frac{(2l)^{2m}}{2^m m!} \le \sum_{m=10^2 l+1}^{\infty} \left( \frac{(2l)^2}{2m/3} \right)^m$$

$$\le \sum_{m=10^2 l+1}^{\infty} \frac{1}{2^m} \le e^{-l}.$$

$\blacksquare$

In light of the above, we use the following notation.

**Definition F.9.** *We will use $\mathcal{P}_l(x)$ to denote the polynomial computed in Lemma F.8 for parameter $l$. Note that $\mathcal{P}_l$ is a polynomial of degree $(10l)^2$ and for $G = N(0,1)$, we have*

$$|G(x) - \mathcal{P}_l(x)| \le e^{-l}$$

*for $x \in [-2l, 2l]$.*

### F.3 Linear Regression

Recall that at the core of the problems we are studying, we are given some function $f$ and want to approximate it as a weighted sum $a_1 f_1 + \cdots + a_n f_n$ of some functions $f_1, \ldots, f_n \in \mathcal{F}$ for some family of functions $\mathcal{F}$. The result below allows us to solve the problem of computing the coefficients if we already know the components $f_1, \ldots, f_n$ that we want to use. The precise technical statement is slightly more complicated in order to incorporate the various types of additional constraints that we may want to impose on the coefficients $a_1, \ldots, a_n$.

**Lemma F.10.** *Let $\mathcal{D}$ be a distribution on $\mathbb{R}$ that we are given. Also assume that we are given functions $f, f_1, \ldots, f_n, g, g_1, \ldots, g_n : \mathbb{R} \to \mathbb{R}$. Assume that there are nonnegative coefficients $a_1, \ldots, a_n$ such that $a_1 + \cdots + a_n \le 1$ and*

$$\int_{-\infty}^{\infty} (f(x) - a_1 f_1(x) - \cdots - a_n f_n(x))^2 \mathcal{D}(x) dx + \int_{-\infty}^{\infty} (g(x) - a_1 g_1(x) - \cdots - a_n g_n(x))^2 \mathcal{D}(x) dx \le \epsilon^2$$

*for some parameter $\epsilon > 0$. Then there is an algorithm that runs in $\mathrm{poly}(n, \log 1/\epsilon)$ time and outputs nonnegative coefficients $b_1, \ldots, b_n$ such that $b_1 + \cdots + b_n \le 1$ and*

$$\int_{-\infty}^{\infty} (f(x) - b_1 f_1(x) - \cdots - b_n f_n(x))^2 \mathcal{D}(x) dx + \int_{-\infty}^{\infty} (g(x) - b_1 g_1(x) - \cdots - b_n g_n(x))^2 \mathcal{D}(x) dx \le 2\epsilon^2 .$$

*Proof.* Let $v = (1, b_1, \ldots, b_n)$. Note that we can write

$$\int_{-\infty}^{\infty} (f(x) - b_1 f_1(x) - \cdots - b_n f_n(x))^2 \mathcal{D}(x) dx + \int_{-\infty}^{\infty} (g(x) - b_1 g_1(x) - \cdots - b_n g_n(x))^2 \mathcal{D}(x) dx = v^T M v$$

where $M$ is a matrix whose entries are $\int_{-\infty}^{\infty} (f(x) f_j(x) + g(x) g_j(x)) \mathcal{D}(x) dx$ in the first row and column and the other entries are $\int_{-\infty}^{\infty} (f_i(x) f_j(x) + g_i(x) g_j(x)) \mathcal{D}(x) dx$. Since all of these functions are given to us, we can explicitly compute $M$. Also note that clearly $M$ is positive semidefinite. Thus, we can compute its positive semidefinite square root, say $N$. Now

$$v^T M v = \|Nv\|_2^2$$

so it remains to solve $\min_v \|Nv\|_2^2$ which is a convex optimization problem that we can solve efficiently (the size of the problem is $\mathrm{poly}(n)$). $\blacksquare$