# OpenReview forum: "Robust Model Selection and Nearly-Proper Learning for GMMs"
_NeurIPS.cc/2022/Conference — NeurIPS 2022 Accept_

### Official Review · Reviewer_27w8 · 2022-06-23

**Rating:** 8
**Confidence:** 4
**Soundness:** 4 excellent
**Presentation:** 4 excellent
**Contribution:** 4 excellent

**Summary:**

This paper addresses the problem of proper learning of 1-d GMMs w.r.t. total variation distance. However, k (number of components) is unknown and we are allowed to output a mixture more than k components (at most polylogarithmic factor more). For non-proper learning, it is already known how to learning 1-d GMMs using piecewise polynomials. Therefore, the main remaining challenge for proper learning is a computational one, not a statistical one. The techniques used to solve the problem are novel and interesting, and compared to the previous work, the number of components of the mixture they find is significantly smaller.

Proof overview: we can assume we have an approximation to the distribution in the form of a piecewise polynomial (based on the previous work). The authors prove that under some conditions (the components are not very thin), there are Gaussian multipliers that can make the mixture "well-conditioned" (concentrated around a mean with similar variances). They then prove that indicator function of every interval can be written as a mixture of Gaussian multipliers. Therefore, the mixture over that interval can be written as the product of the mixture and the indicator. If the interval is such that the conditions for thin Gaussians and the multipliers are met, then the product is a well-conditioned mixture. They then prove that there are intervals that satisfy the condition and most of the mass of mixture is concentrated on those intervals.

**Questions:**

no questions at this stage

**Limitations:**

I don't foresee direct negative impact of this theoretical work

**Strengths And Weaknesses:**

Strengths:

+ The paper addresses an important and basic problem in the area of density estimation: proper learning of 1-d GMMs
+ There have been multiple previous attempts on this problem; but this paper significantly improves over them
+ The algorithmic techniques as well as the structural results are novel and interesting
+ The presentation is nice

Weaknesses (minor issues):
+ Sometimes the authors jump in to the technical steps of the proofs without explaining the big picture. Adding a few sentences at the beginning of these subsections can help readability. For example, in the localization section, it is not clear from the beginning that the localization is going to be centered at different components (The reader may assume it is going to be a single localization for the whole mixture). I was not able to understand the techniques thoroughly until I read the instructions in the appendix.
+ Missing discussion of a seemingly very relevant paper by X. Wu and C. Xie, “Improved algorithms for properly learning mixture of gaussians”.

---

> ### Author Response · Authors · 2022-08-02
> **Response to Reviewer 27w8**
>
> We will revise the presentation to include more high-level descriptions for each subsection.  We will also add a discussion about the paper by [Wu and Xie 2018].  We note that this paper outputs a mixture with $poly(k/\epsilon)$ components and thus falls into category (4) in the discussion in the introduction.

---

### Official Review · Reviewer_8UxQ · 2022-07-07

**Rating:** 5
**Confidence:** 4
**Soundness:** 3 good
**Presentation:** 3 good
**Contribution:** 2 fair

**Summary:**

This paper considers density estimation of 1-D Gaussian mixtures with unknown number of components $k$. Main point of this submission is to find a 1-D Gaussian mixture hypothesis with at most $\tilde{O} (k) = k \cdot {\rm poly} \log(1/\epsilon)$ components, such that the returned hypothesis is $O(\epsilon)$-close in TV distance to the ground truth. Previous work on the same problem used either larger number of components $O(1/\epsilon^2)$ or used non Gaussian-mixture hypothesis. Main tools used here are polynomial approximation of Gaussian p.d.f. up to degree $O(\log^2 (1/\epsilon))$. It is shown that the techniques shown for 1-D Gaussian mixture can also be used for sparse Fourier reconstruction.

In my understanding, the proposed algorithm proceeds in two steps: (1) run any improper learning algorithm that outputs a GMM hypothesis with much more number of components (the authors' pick is [Chan et al., 2013] that uses $\tilde{O}(k/\epsilon^2)$ samples) and (2) reduce the number of used components using polynomial approximation of Gaussian p.d.f s. Main subtlety in processing (2) is that if variances (of individual Gaussians) across different components vary a lot, then it is trickier to find polynomial functions that can approximate all p.d.f. s of Gaussian components. Authors proposed a "localization technique" to handle this.

**Questions:**

1. About title: I feel that the word 'robust' in title is slightly misleading, because this paper is not really about handling corruptions.

2. Also, I am not sure 'nearly-proper' is a well-defined terminology to use. We don't know whether $k \cdot {\rm poly} \log (1/\epsilon)$ is the best we can do. Do you have any thoughts on whether O(k) components without log factors possible?

3. The crux of the result is to apply Caratheodory's theorem in polynomial function space. This sounds fairly standard. Could you elaborate on how standard or novel this technique is in density estimation literature?

**Limitations:**

I don't see any negative societal impact.

**Strengths And Weaknesses:**

*Strength*

- Presentation of the paper is very clear. It would be even nicer if you have an algorithm box or enumeration block that summarizes the full algorithm.

- The paper also provides a nice application (sparse Fourier reconstruction) of their theoretical findings.


*Weakness*

- Techniques used in this paper are very specific to the 1-D GMM case, and thus significance of the results is limited.

- The paper is mostly about the post-processing to reduce the number of components. I wonder if there is anything improved in the density estimation step compared to benchmarks (e.g., Li and Schmidt, 2017).

- If there is any interesting challenge that did not appear in 1-D GMM but should be considered in the sparse Fourier problem, then it might be better fleshed out in the main text.

---

> ### Author Response · Authors · 2022-08-02
> **Response to Reviewer 8UxQ**
>
> Our paper indeed only considers the case of 1D GMMs.  However, there is a large body of work on the 1D case and still there are very fundamental questions that remain open.  As discussed in the introduction, learning a GMM with the minimum number of components is important for model selection and is a key part of understanding the underlying distribution.  Our work makes significant progress towards this goal, efficiently outputting a GMM with the minimum number of components up to a logarithmic factor whereas previous algorithms either require exponential runtime or significantly more components.
>
>
> The crux of our algorithm is post-processing a density estimate (e.g. from [Chan et al., 2013]) as the density estimation guarantees obtained there are already essentially optimal.  Since the known density estimation guarantees are nearly optimal, the later works about proper learning e.g. [Li and Schmidt, 2017] are also purely about post-processing a density estimate.  However, the runtime that they require is exponential in $k$.
>
>
> The term near-proper is not standard.  However, we think it is important to emphasize that our approach computes the minimum number of components up to logarithmic factors.  In terms of getting fully proper learning, we suspect that eliminating the $\log 1/\epsilon$ dependence is computationally hard. It is equivalent to finding a non-trivially sparse solution to a system of polynomial equations and there seems to be no structure that makes algorithmic search better than brute-force possible. This question has already been open for many years, but there hasn't been any progress on proper learning. Hence we introduced the near-proper version as a new way to make inroads.  We will include further discussion about this in the next version.
>
>
> Why do we call our algorithms robust? Our results hold when we obtain samples from any distribution that is $\epsilon$-close in TV distance to a GMM with $k$ components.  This is equivalent to being able to handle an $\epsilon$-fraction of corruptions from an oblivious adversary and hence we use the term robust.  Note that many approaches, e.g. those based on MLE are not naturally robust.  In terms of techniques, indeed there are previous works that use similar expansions in polynomial space to prove that an approximation with $\log 1/\epsilon$ components exists when the mixture is well-conditioned e.g. [Polyanskiy and Wu 2020] (our approach is slightly different because it needs to be algorithmic and robust).  Moreover, to deal with general mixtures, we need several additional ingredients.  In particular, we use a carefully chosen localization scheme to reduce the problem to learning several well-conditioned submixtures.  We then show how to reconstruct the full distribution from these localizations.

---

### Official Review · Reviewer_r5gD · 2022-07-11

**Rating:** 7
**Confidence:** 4
**Soundness:** 4 excellent
**Presentation:** 4 excellent
**Contribution:** 3 good

**Summary:**

In learning theory, we often assume that samples are drawn from a finite mixture model.
Also, the number of components in the finite mixture model is known.
However, in many applications, this number may not be known in advance.
The problem of finding the number of components is called model selection.
The authors study the problem of robust model selection for one dimensional Gaussian mixture models with $k$ components ($k$-GMM).
Formally, it is formulated as follows.
Given samples drawn from a distribution that is $\epsilon$-close to a $k$-GMM in total variation, we want to output a $k'$-GMM where $k'$ is not much larger than $k$.

There is a long line of existing works for learning GMM.
However, they have different shortcomings in this setting such as requiring exponentially (in $k$) many samples, running in time in exponential in $k$, the number of Gaussians in the output $k'$ much larger than $k$, etc.
In this paper, the authors give an algorithm for learning arbitrary one dimensional $k$-GMMs that outputs an approximation with $\tilde O(k)$ components.

The algorithm works as follows.
Recall that we are given samples drawn from a distribution $f$ and there is a $k$-GMM $\mathcal{M}$ $\epsilon$-close to $f$ in total variation.
The authors first consider the following case.
When the underlying GMM $\mathcal{M}$ satisfies the property that the means of the Gaussians are all not too far away and the variances of the Gaussians are all constant scale, then we can find a $O(\log \frac{1}{\epsilon})$-GMM $\widetilde{\mathcal{M}}$ that is also close to $f$ in total variation.
It is what the authors call well-conditioned.
Since the Gaussians are well-conditioned, achieving this is rather easy by examining the Taylor expansion.
Then, the authors reduce the general case to the well conditioned case via localization.
The high level idea is to look for the intervals such that the union of these intervals covers the union of the significant intervals of the Gaussians in $\mathcal{M}$ where the significant interval of a Gaussian is a long interval covering most probability mass of this Gaussian.
For each interval $I$, the mixture of Gaussians in $\mathcal{M}$ whose significant interval intersects $I$ forms a mixture of well conditioned Gaussians.
On the other hand, the Gaussians in $\mathcal{M}$ whose significant interval does not intersect this interval $I$ are negligible with respect to this mixture of well conditioned Gaussians.
Hence, the authors use dynamic programming to find out these intervals from the samples and use the algorithm for the well conditioned case to output a mixture of $\widetilde O(k)$ Gaussians since the number of these intervals is $\widetilde O(k)$ and there are $\log \frac{1}{\epsilon}$ Gaussians in the mixture for the well conditioned case.

Furthermore, the authors apply the result to solve hypothesis testing for model selection problem and the techniques to solve sparse Fourier problem.



**Questions:**


Questions:
- Is there a sample complexity lower bound showing that no efficient algorithm outputs a mixture of $k\cdot N$ Gaussians where $N$ is sub-poly-logarithmic of $\frac{1}{\epsilon}$ or even independent of $\frac{1}{\epsilon}$ under this setting?

Minor note:
- In line 219, what is $G_0$?

**Strengths And Weaknesses:**


Strengths:
- The paper is well-written.
The ideas and the proofs are easy to follow.
- The results is interesting.
The number of Gaussians in the output is $k(\log \frac{1}{\epsilon})^{O(1)}$ while this number is $k(\frac{1}{\epsilon})^{O(1)}$ in the previous works.
This improvement for the dependence on $\frac{1}{\epsilon}$ from a polynomial bound to a logarithmic bound is interesting.


Weaknesses:
- It may just be a minor weakness.
In terms of proof techniques, it seems that the techniques used in this paper are rather standard.
It seems the tools used in the proof are not very surprising.

---

> ### Author Response · Authors · 2022-08-02
> **Response to Reviewer r5gD**
>
> There is no sample complexity lower bound as it is information-theoretically possible to brute-force search over all possible mixtures of Gaussians to find the one with the minimum number of components that fits the data.  The problem that we study is computational, as we focus on efficient algorithms for fitting a mixture with as few components as possible.  In this regard, we suspect that eliminating the $\log 1/\epsilon$ dependence is computationally hard. It is equivalent to finding a non-trivially sparse solution to a system of polynomial equations and there seems to be no structure that makes algorithmic search better than brute-force possible. This question has already been open for many years, but there hasn't been any progress on proper learning. Hence we introduced the near-proper version as a new way to make inroads.  We will include further discussion about this in the next version.
>
>
> In terms of techniques, while Taylor expansions have been used before in the context of learning GMMs, we believe that the use of localization is novel.  Furthermore, we emphasize that the localization must be done very carefully to localize around thin (small variance) components first as these components can drastically affect whether a sub-mixture is well-conditioned.  In line 219, we meant for $G_0 = N(\mu, \sigma^2)$.  We will fix this in the next version.

---

### Official Review · Reviewer_bUkB · 2022-07-12

**Rating:** 8
**Confidence:** 3
**Soundness:** 4 excellent
**Presentation:** 4 excellent
**Contribution:** 4 excellent

**Summary:**

The paper considers the fundamental problem of model selection for univariate Gaussian Mixture Models (GMMs), which asks for the minimum number of components needed to fit the given data. The approach to do this is via robust proper learning of GMMs, where given access to a distribution that is $\epsilon$-close to a $k$-GMM we want to find a $k’$-GMM (having ideally $k’=k$) being $\tilde{O}(\epsilon)$-close to that distribution. The paper presents the first algorithm for this task with $k’=\tilde{O}(k)$ that works for arbitrary GMMs and has polynomial sample complexity and runtime. Using that result, it obtains an algorithm for the model selection problem. Finally, the paper utilizes the techniques to obtain a result for Fourier sparse interpolation that enjoys better bounds on the sparsity of the outputted signal than prior work. At the core of the GMM algorithm is a procedure that works for GMMs with well-conditioned components and a localization technique allowing to reduce the general case to the well-conditioned one. The former is based on considering the Taylor expansions of the Gaussian components and associating the well-conditioned mixtures to points in a convex body. The latter uses Gaussian multipliers to reweight the mixtures so that they become more localized and constructs $\tilde{O}(k)$ such multipliers that allow reconstruction of the original mixture.

**Questions:**

One question for the authors is if the localization technique or something similar has also been employed in prior work for GMMs or other problems.

**Limitations:**

Sufficiently addressed.

**Strengths And Weaknesses:**

Questions regarding learning of Gaussians and mixture models are arguably one of the most fundamental problems in statistics. The paper presents an algorithm that enjoys polynomial runtime while working under no additional assumptions on the input.
The paper is technically interesting and employs an interesting suite of techniques. The solution deviates from prior work ideas and requires substantial technical work.
The paper is also well-written and despite the fact that the arguments used are involved, the authors managed to present their ideas in a clean way. I have not carefully checked the details in the supplementary material, but the outline of the main submission seemed reasonable.

I recommend that the paper be accepted.

---

> ### Author Response · Authors · 2022-08-02
> **Response to Reviewer bUkB**
>
> Our use of localization is related to uses of localization in compressed sensing and sparse Fourier transforms where one multiplies the Fourier transform by some kernel to localize it.  We are not aware of uses of localization in the context of GMMs.  Our use of localization is also different as we crucially use a Gaussian kernel instead of a compactly supported kernel to ensure that after localizing, the components remain Gaussian.  This complicates the reconstruction step but nevertheless, we are able to reconstruct the mixture from the localized pieces.

---

### Meta-Review · Area_Chair_gQBs · 2022-08-28

**Recommendation:** Accept
**Confidence:** Certain

**Metareview:**

This paper gives significantly improved guarantees for (im)proper learning of Gaussian mixtures in 1-dimension. Given samples from a distribution that is close to a mixture of Gaussians, it gives an polynomial time (and samples) algorithm that outputs a mixture with (slightly) more components that is also close. The paper's contributions are solid both in terms of the result and the techniques involved. So this is a clear accept.

**Award:**

No

---

### Decision · Program_Chairs · 2022-09-14

Accept